# Systematic single-variant and gene-based association testing of thousands of phenotypes in 394,841 UK Biobank exomes

## Graphical abstract

## Authors

Konrad J. Karczewski,
Matthew Solomonson,
Katherine R. Chao, ..., Heiko Runz,
Melissa R. Miller, Benjamin M. Neale

## Correspondence

konradk@broadinstitute.org (K.J.K.),
bneale@broadinstitute.org (B.M.N.)

## In brief

Karczewski et al. generated a massive-scale association dataset between rare genetic mutations and thousands of diseases and traits and released these data in the Genebass browser. They quantify the influence of natural selection and gene function on association discovery and highlight an association between *SCRIB* and a brain-imaging trait.

## Highlights

- Public release of gene-based association statistics for 4,529 diseases and traits

- Genebass, a browser framework to display rare-variant associations

- Tight coupling between frequency, natural selection, and power for genetic discovery

- Biological signal between *SCRIB* and white-matter integrity (from MRI)

 Karczewski et al., 2022, Cell Genomics 2, 100168
September 14, 2022 © 2022 The Author(s).

CellPress

## Resource

# Systematic single-variant and gene-based association testing of thousands of phenotypes in 394,841 UK Biobank exomes

Konrad J. Karczewski,[1,2,3,11,13,*] Matthew Solomonson,[1,2,11] Katherine R. Chao,[1,2,11] Julia K. Goodrich,[1,2] Grace Tiao,[1,2] Wenhan Lu,[1,2,3] Bridget M. Riley-Gillis,[4] Ellen A. Tsai,[5] Hye In Kim,[6] Xiuwen Zheng,[4] Fedik Rahimov,[4] Sahar Esmaeeli,[4] A. Jason Grundstad,[4] Mark Reppell,[4] Jeff Waring,[4] Howard Jacob,[4] David Sexton,[5] Paola G. Bronson,[5] Xing Chen,[6] Xinli Hu,[6] Jacqueline I. Goldstein,[1,2,3] Daniel King,[1,2,3] Christopher Vittal,[1,2,3] Timothy Poterba,[1,2,3] Duncan S. Palmer,[1,2,3] Claire Churchhouse,[1,2,3] Daniel P. Howrigan,[1,2,3] Wei Zhou,[1,2] Nicholas A. Watts,[1,2] Kevin Nguyen,[1,2] Huy Nguyen,[1,2] Cara Mason,[7] Christopher Farnham,[7] Charlotte Tolonen,[7] Laura D. Gauthier,[7] Namrata Gupta,[7] Daniel G. MacArthur,[1,2,9,10] Heidi L. Rehm,[1,2] Cotton Seed,[1,2,3] Anthony A. Philippakis,[7] Mark J. Daly,[1,2,3,8] J. Wade Davis,[4,12] Heiko Runz,[5,12] Melissa R. Miller,[6,12] and Benjamin M. Neale[1,2,3,*]

[1]Program in Medical and Population Genetics, Broad Institute of MIT and Harvard, Cambridge, MA 02142, USA
[2]Analytic and Translational Genetics Unit, Massachusetts General Hospital, Boston, MA 02114, USA
[3]Stanley Center for Psychiatric Research, Broad Institute of MIT and Harvard, Cambridge, MA 02142, USA
[4]Genomics Research Center, AbbVie, North Chicago, IL 60064, USA
[5]Biogen, Inc., Cambridge, MA 02142, USA
[6]Worldwide Research Development and Medical, Pfizer, Inc., Cambridge, MA 02139, USA
[7]Data Sciences Platform, Broad Institute of MIT and Harvard, Cambridge, MA 02142, USA
[8]Institute for Molecular Medicine Finland, Helsinki, Finland
[9]Present address: Center for Population Genomics, Garvan Institute of Medical Research and UNSW, Sydney, NSW, Australia
[10]Present address: Murdoch Children's Research Institute, Parkville, VIC, Australia
[11]These authors contributed equally
[12]These authors contributed equally
[13]Lead contact
*Correspondence: konradk@broadinstitute.org (K.J.K.), bneale@broadinstitute.org (B.M.N.)

## SUMMARY

Genome-wide association studies have successfully discovered thousands of common variants associated with human diseases and traits, but the landscape of rare variations in human disease has not been explored at scale. Exome-sequencing studies of population biobanks provide an opportunity to systematically evaluate the impact of rare coding variations across a wide range of phenotypes to discover genes and allelic series relevant to human health and disease. Here, we present results from systematic association analyses of 4,529 phenotypes using single-variant and gene tests of 394,841 individuals in the UK Biobank with exome-sequence data. We find that the discovery of genetic associations is tightly linked to frequency and is correlated with metrics of deleteriousness and natural selection. We highlight biological findings elucidated by these data and release the dataset as a public resource alongside the Genebass browser for rapidly exploring rare-variant association results.

## INTRODUCTION

Coding variation has been the most readily interpretable class of genomic variation since the development of the gene model and mapping of the human genome. As such, it has facilitated the mapping and interpretation of variants with immediate clinical importance such as the American College of Medical Genetics actionable variant list.[1] More recently, exome sequencing has yielded the discovery of specific causal variants for hundreds of rare diseases, particularly dominant acting *de novo* variants for severe diseases.[2]

As the sample sizes of exome sequencing datasets continue to grow, so do the opportunities to identify associations between rare variants and phenotypes (both complex traits and diseases). In complex diseases, identifying causal genetic factors for a given disease can provide direct insight into the potential for therapeutic avenues. For instance, gain-of-function variants in *PCSK9* have been demonstrated to increase low-density lipoprotein (LDL) levels and thus risk for cardiovascular disease.[3] Accordingly, loss-of-function (LoF) variants are protective for cardiovascular disease,[4] and less than 15 years after the discovery of this effect, therapeutic approaches to inhibit PCSK9 have been brought to market.[5]

Deeply phenotyped biobanks present a unique opportunity to simultaneously analyze multiple diseases and traits within a single cohort, enabling the discovery of new disease genes with

therapeutic potential at a large scale, such as the identification of rare variants in *ANGPTL7* that protect against glaucoma.[6] The UK Biobank is a collection of approximately 500,000 participants with standardized, detailed phenotypic data[7] on which genome-wide association studies (GWASs) have been run extensively. The UKB Exome Sequencing Consortium, a partnership between the UKB and 8 biopharma companies, generated exome sequences for this cohort,[8] and recent studies have used the exome sequence data to explore various aspects of rare-variant associations, including novel biological signals for type 2 diabetes[9] and cardiometabolic traits,[10] as well as cross-phenotype analyses that identify new hits for a variety of traits.[11–13] Here, we describe results from a systematic, large-scale rare-variant association analysis of 4,529 phenotypes, release these full sets of summary statistics in a results browser, and explore the role of natural selection and allele frequency on rare-variant associations.

## RESULTS

### Generating high-quality exome data for rare-variant associations

We built an end-to-end pipeline for read mapping, processing, joint variant calling, quality control (QC), and mixed-model association analysis and applied this pipeline to 454,697 individuals with exome sequence data from the UK Biobank. The read mapping and processing pipeline adopted the GATK Best Practices pipeline (GRCh38), and the resulting variants (gVCF files) were joint called using a scalable implementation in Hail (Figure S1).[14] We processed a set of 4,529 phenotypes including 1,233 quantitative traits as well as 3,296 binary traits with at least 200 cases, which included 725 disease endpoints based on Internal Classification of Diseases (ICD)-10 codes (Figure S2).

After performing QC in a similar but augmented (e.g., array concordance; see Supplemental information) manner as for the Genome Aggregation Database (gnomAD),[15] we generated a high-quality dataset of 450,953 individuals (Figures S3–S5; Table S1) including related individuals. This included 394,841 individuals of European ancestry in which we find 23,880,790 high-quality variants (Figure S6). For each of the 19,407 protein-coding genes, we considered up to four functional annotation categories: predicted LoF (pLoF), missense (including low-confidence pLoF variants and in-frame insertions or deletions [indels]), synonymous, and the combination pLoF or missense group, resulting in 8,074,878 variants and 75,767 groups for association testing (i.e., one group per gene and functional annotation category).

### Creating a high-quality set of rare-variant associations

We performed group tests using the mixed-model framework SAIGE-GENE,[16] which includes single-variant tests and gene-based burden (mean), SKAT (variance), and SKAT-O (hybrid variance/mean) tests (Figure S7). In total, we performed up to 8,074,878 single-variant tests and 75,767 group tests for each of 4,529 phenotypes (Figure 1). Additionally, we generated 314 heritable random phenotypes to test the asymptotic properties of the mixed-model association testing framework (Figures S8 and S9) and to determine empirical p value thresholds for Type I error control. Based on this analysis, for each phenotype, in

addition to QC criteria defined below, we consider genome-wide p value thresholds of $2.5 \times 10^{-7}$ for SKAT-O tests, $6.7 \times 10^{-7}$ for burden tests, and $8 \times 10^{-9}$ for single-variant tests (see Supplemental information and Figure S10), corresponding to approximately 0.05 expected false positives per phenotype.

We performed extensive QC on these summary statistics (Figure 1; Table S2), including a minimum of two variants per group test, a minimum coverage of 20×, and a minimum expected allele count (frequency × n_cases) of 50 for the summary statistics, respectively, as well as genomic control (lambda GC) for each phenotype and each gene (Figures S11–S15). Further, we pruned to a set of 3,819 high-quality independent phenotypes encompassing 677 continuous traits and 3,142 binary traits, including 708 ICD codes (Figures 1A and S16; Table S2). We confirmed the robustness of our results by comparing them wiht a previous large-scale study of height (Tables S3–S5; Figure S17) and red blood cell phenotypes (Table S6), for which our analysis replicates the majority of associations with consistent direction of effect.[17,18]

We filtered to 263,696 variants, including 6,117 pLoF variants, 155,705 missense variants, and 101,874 synonymous variants with at least one phenotype having an expected allele count (cohort frequency × n_cases) over 50 (Figure 1B). For group tests, we filter to a high-quality set of 54,647 gene tests with at least 20× coverage (Figure S13) and at least one phenotype with expected allele count ≥50 for pLoF (7,296 genes), missense (15,943), synonymous (16,014), and pLoF or missense (15,394) (Figure 1C).

Using these criteria, we identified a total of 68,623 and 6,403 associations meeting our p value threshold with a mean of 18.0 and 1.7 associations per phenotype for single-variant tests and group tests, respectively (disease results shown in Figures 2A and 2B). Comparing the group test results with single-variant association test results, we find that single-variant tests identify more significant associations than group tests, as these are largely from common variants that are excluded from the group tests. However, we also find 1,849 associations (on average 0.48 per phenotype) from group tests where no single-variant association reached our p value threshold for any single variant in the corresponding gene (Figure 2C). Further, most associations arise from missense and synonymous variants, as expected from their greater numbers in the exome, particularly from single-variant associations. However, pLoF variants exhibit relatively more associations in group tests, which is consistent with these variants being individually rare but directionally consistent, resulting in increased power in a group test (Figure 2D). In combined tests of pLoF and missense variants, we find an additional 275 associations among burden tests (254 for SKAT-O) that are significant for the combined test but not missense or pLoF tests alone.

### Displaying rare-variant associations

The utility of human genetic variation datasets are substantially enhanced when made accessible in the form of online portals that enable non-technical domain experts to quickly browse, interpret, and export results for downstream follow up.[19] We extended our gnomAD browser toolkit to create the Genebass (gene-biobank association summary statistics) browser (https://genebass.org), a new, highly interactive tool for exploring large numbers of

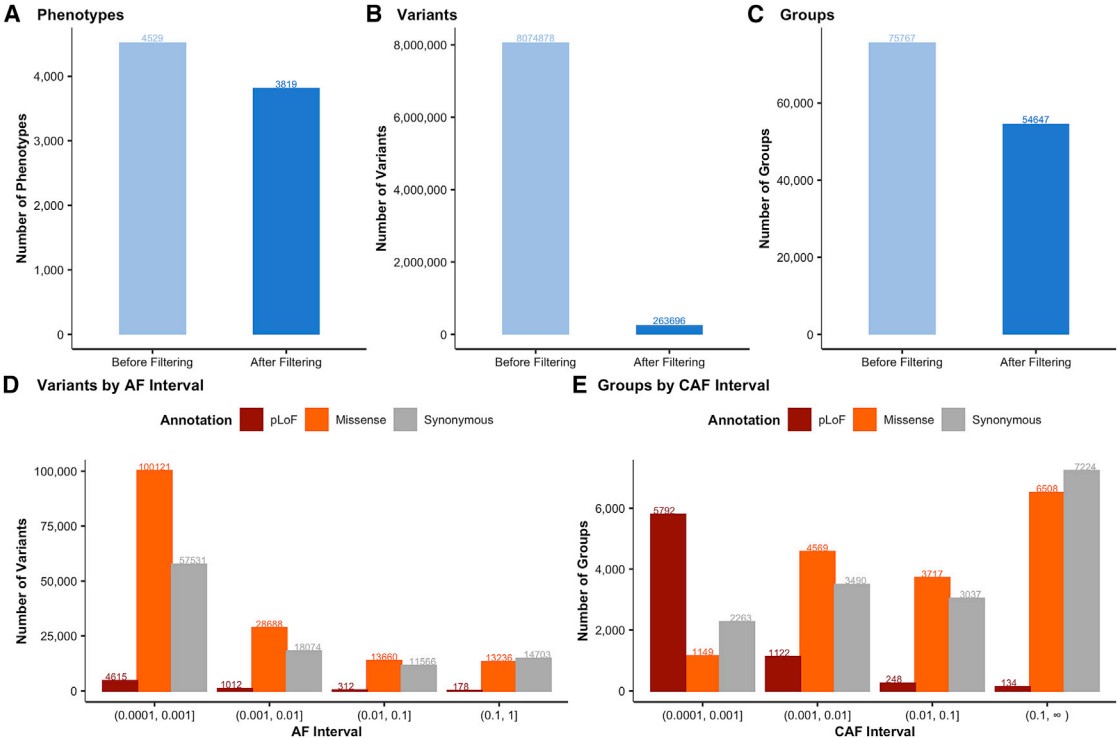

**Figure 1. Quality control (QC) of rare-variant association tests**

(A–C) The number of phenotypes (A), variants (B), and groups (i.e., gene-annotation pairs; C) before and after QC.
(D and E) After QC, the number of variants (D) and genes (E) are broken down by annotation and frequency bin (alternate allele frequency [AF] for variants, cumulative AF [CAF] for genes).

gene-based phenome-wide association study (PheWAS) analysis results. This resource provides users with direct access to all 4,529 phenotypes, serving up 993,280,477 gene-level association statistics (across 19,407 genes, 4 annotation sets, and 3 burden tests) and 28,158,190,538 single-variant association statistics across 8,074,878 exome variants. For completeness, the released dataset includes all association statistics, including pre-QC data, but we provide functionality to filter to only the highest quality data presented herein. Our web application features a unique layout and navigational scheme for rapidly browsing phenome-wide associations by integrating results across genes and variants. Customizable controls, plots, and tables enable flexible filtering and visualization of phenotypes, genes, and variants of interest, results can be exported for downstream analyses, and variant associations across traits can be compared with inform pathways associated with complex traits and develop therapeutic hypotheses (see Supplemental information).

### Frequency and selection affect the landscape of rare-variant associations

The relationship between natural selection, allele frequency, effect size, and power for discovery is a major complexity in the analysis and interpretation of association statistics, particularly from rare variants. The power to detect association is proportional to the variance explained of a bi-allelic variant.[20] Specifically, for a continuous trait, the variance explained of a bi-allelic variant that is purely additive is $2pqa$,[2] where $p$ is the allele fre-

quency, $q = 1-p$, and $a$ is the allelic effect of the variant. Thus, for a fixed effect size, a more common variant will capture more variance and, by extension, show stronger association.

However, the process of negative selection will tend to decrease the frequency of functional damaging variants, suggesting that variants with large effect sizes are more likely to be rare. Indeed, partitioned heritability analyses for common variants support the presence of these countervailing forces, as comparatively lower frequency variants have larger absolute effect sizes, but this growth in effect size is slower than the loss in variance explained from their lower frequency.[21] In evaluating the landscape of rare-variant association, we observe a similar pattern with increasing proportion of variants associated with at least one phenotype as frequency increases (Figure 3A). However, within each frequency category, we observe the effect of functional annotation, a known correlate for deleteriousness, on the association statistics.

Comparing the number of associations by variant annotation in each allele frequency category, we find that pLoF variants have a larger number of associations than missense variants, followed by synonymous variants for single-variant tests (Figure 3A) as well as group tests (Figure 3B). For common variants (>1%), we observe further increases in associations due to power, but with attenuated associations for pLoF variants, likely due to an increased rate of artifacts at common pLoF variants[22] (Figure S18). Within missense variants, variant deleteriousness as predicted by PolyPhen2[23] is correlated with the number of associations

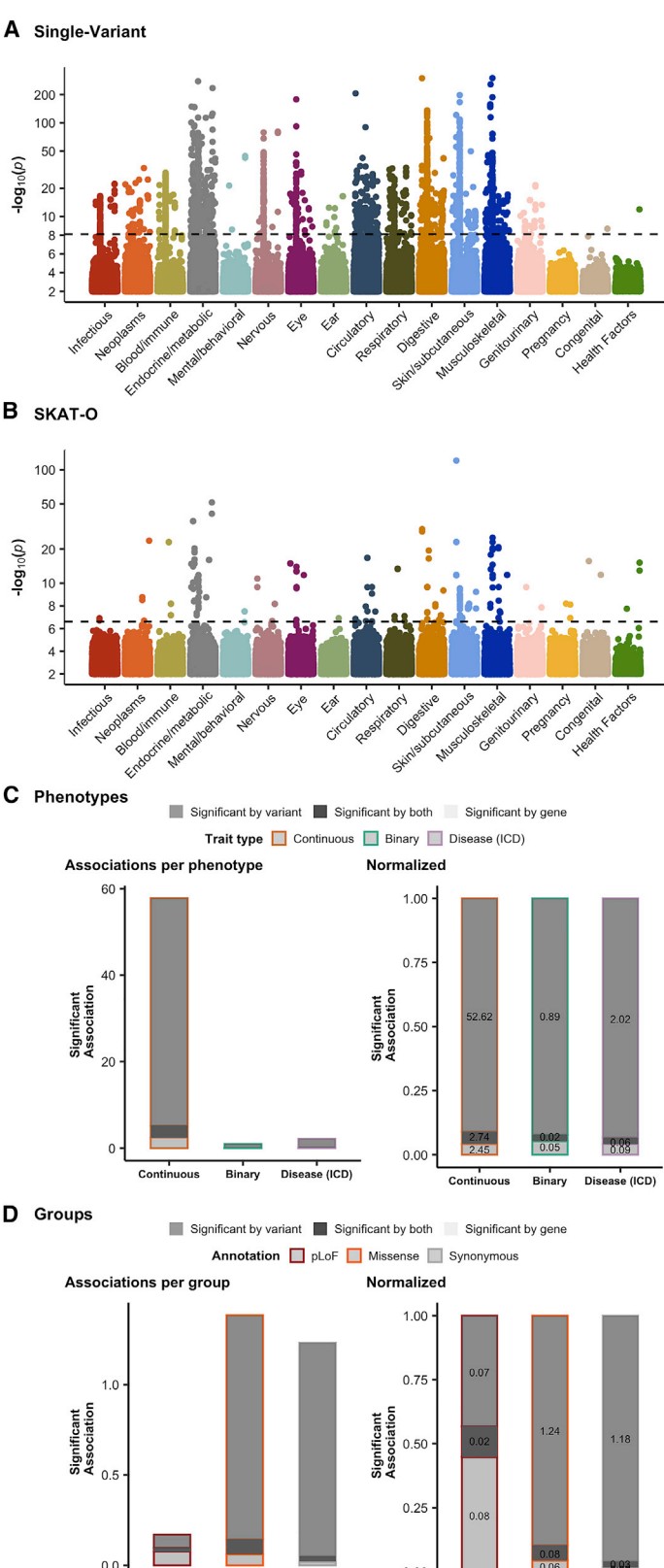

**Figure 2. Rare-variant association testing is enhanced by group tests**

(A and B) For each ICD chapter, we show a Manhattan plot, depicting the distribution of p values for all single-variant (A) and SKAT-O gene-based (B) associations, where for each variant/gene, the minimum p value across phenotypes within each category is shown.

(C and D) The number of gene-level associations per phenotype is shown as a bar plot, broken down by trait type (left) and normalized within each trait type (right), separated by phenotype category (C) or functional annotation (D). The single-variant tests are grouped into genes where at least one associated variant is necessary to be "significant by variant," which is shown alongside group tests ("significant by gene") as well as genes where an association is found both for group and single-variant tests.

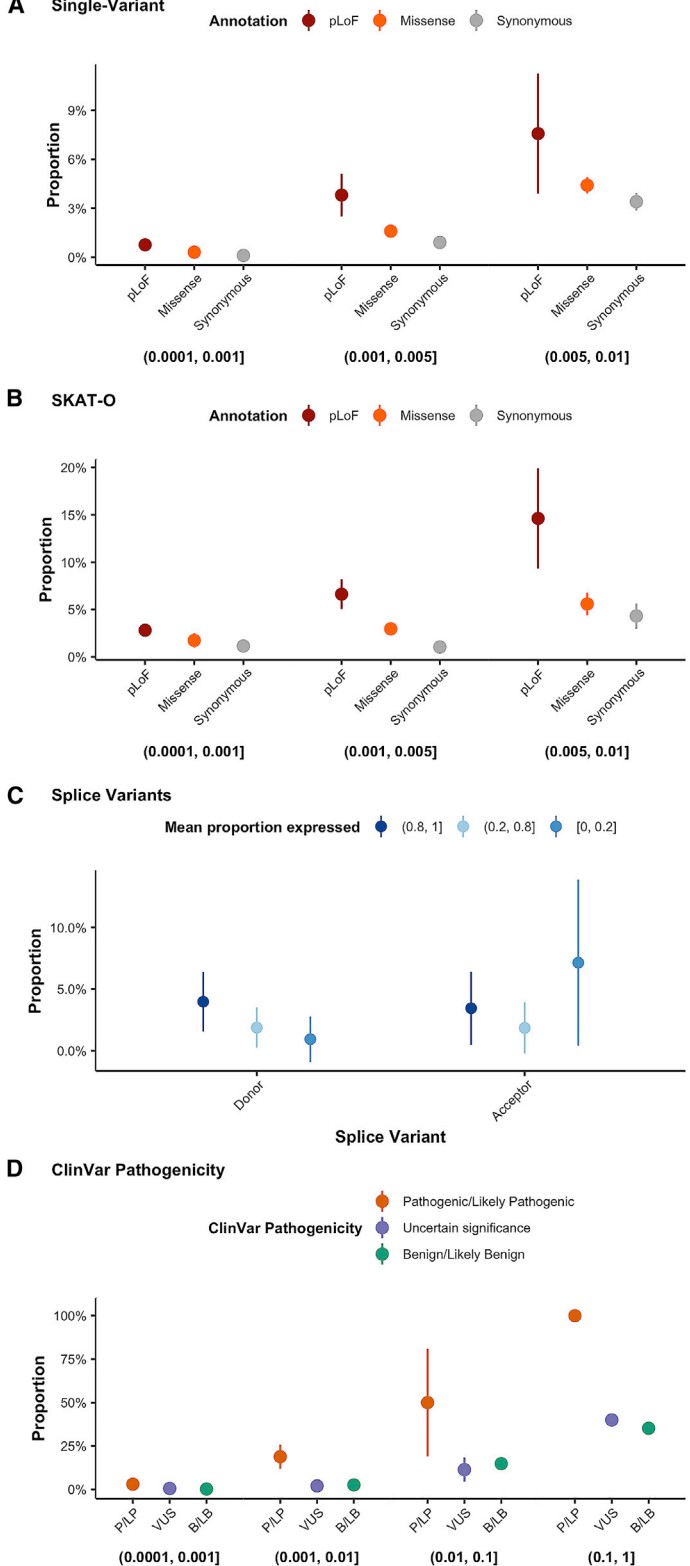

**A** Single-Variant

**B** SKAT-O

**C** Splice Variants

**D** ClinVar Pathogenicity

**Figure 3. The influence of variant AF and functional annotation in exome-association testing**

(A and B) The proportion of single variants (A) and genes (B) with at least one significant hit is shown broken down by AF (A) or CAF (B) category, each shown below the plot, broken down by functional annotation.

(C and D) This metric is also plotted by the proportion expressed across transcripts for splice variants (C) and ClinVar pathogenicity status (D). Error bars represent 95% confidence intervals.

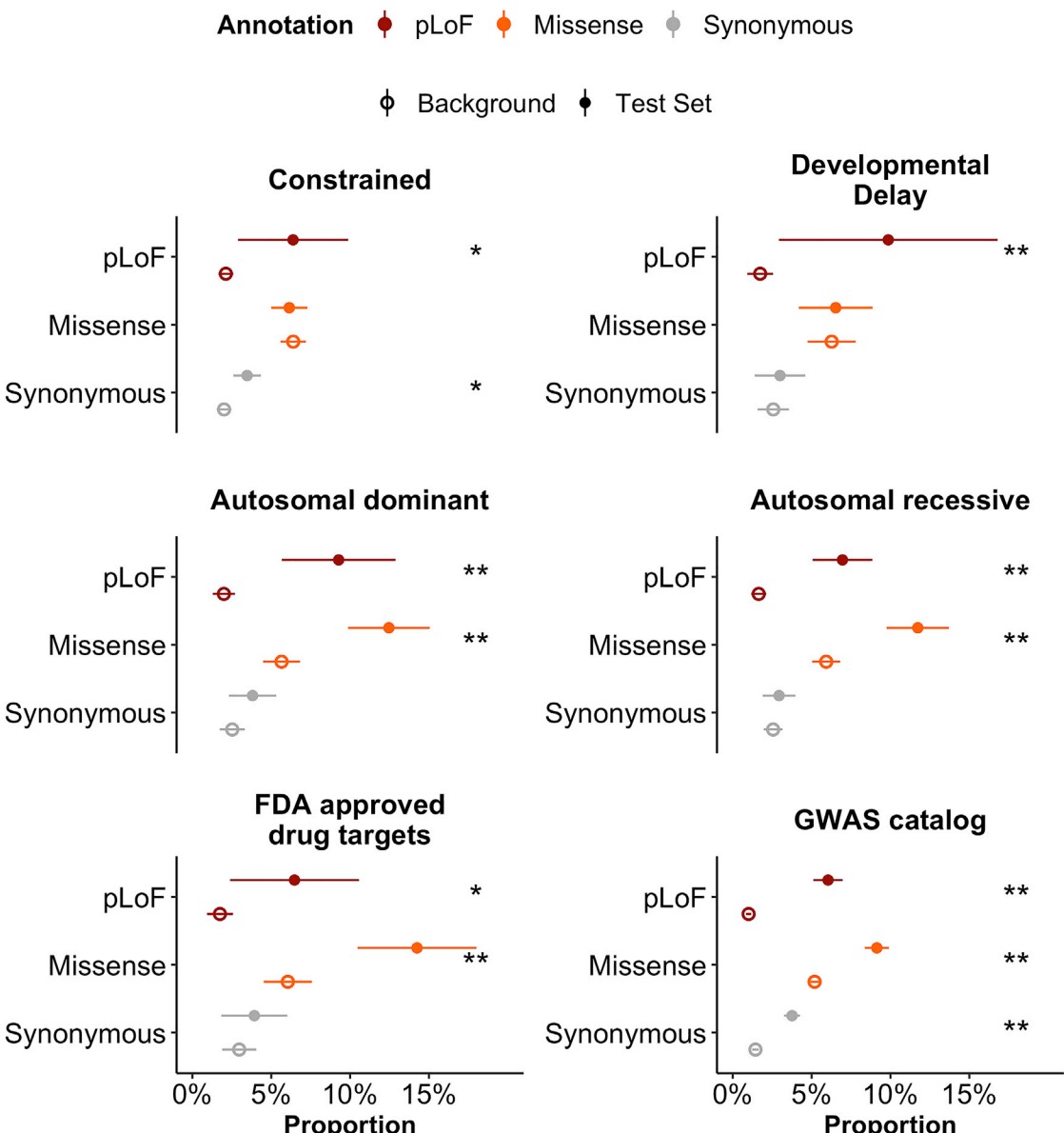

**Figure 4. The effect of gene function on the landscape of rare-variant associations**
The proportion of gene-annotation pairs with at least one association (SKAT-O p < 2.5 × 10⁻⁷) is shown for a number of gene categories, each compared with a background set of genes matched on CAF. Error bars represent 95% confidence intervals. Asterisks denote a significant difference between the background and test sets (*p < 0.05 and **p < 0.001, respectively).

meeting our p value threshold (Figure S18). For splice donor variants, we find a correlation between the proportion expressed across transcripts (pext)[24] and the number of associations (Figure 3C). Additionally, the pathogenicity level of ClinVar variants is correlated with phenotypic association (Figure 3D).

**Gene function influences association statistics**
We examined the phenotypic impact of gene categories previously known to have functional relevance and/or a role in disease. In particular, we find that 470 genes previously implicated in developmental delay[25] are more likely to be associated with

a phenotype in the UK Biobank (Fisher's exact p = 6.6 × 10⁻⁴, odds ratio [OR] = 6.16; Figure 4) than a frequency-matched background set of genes. Further, we observe a correlation between selection against pLoFs in a gene and the phenotypic impact of pLoFs in that gene: specifically, constrained genes (i.e., those in the highest decile of LoF observed/expected upper bound fraction [LOEUF], a metric of LoF intolerance) are more likely to be associated with a phenotype (6.38%) than a frequency-matched set of genes (2.12%; Fisher's exact p = 1.2 × 10⁻³, OR = 3.14; Figure 4). Similarly, genes with known autosomal dominant and autosomal recessive diseases, as

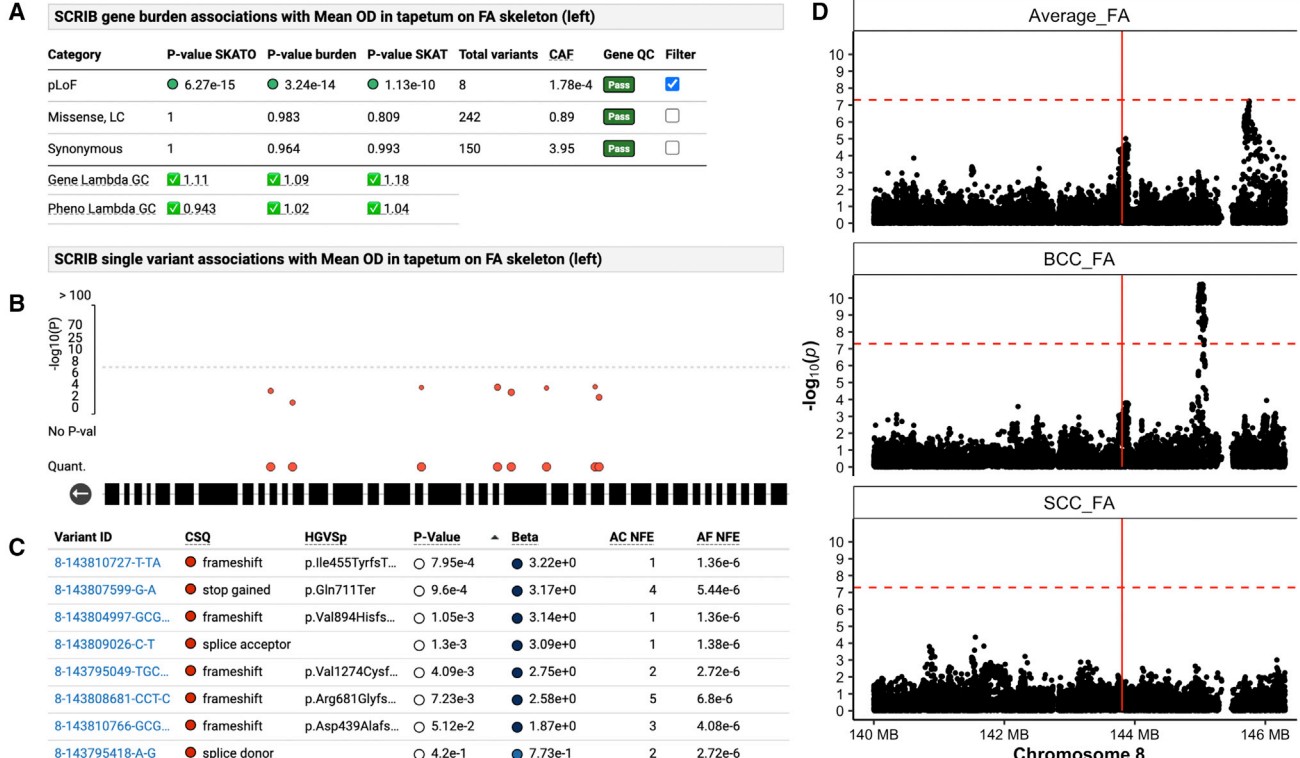

**Figure 5. Refined association between SCRIB and white-matter integrity of tapetum**

The Genebass browser provides views of the full dataset, including all quality control metrics and association statistics.

(A) The summary of association information between pLoF variants in SCRIB with mean orientation dispersion (OD) index in tapetum on fractional anisotropy (FA) skeleton (from diffusion magnetic resonance imaging [dMRI] data).

(B) A rare variant Manhattan plot of 8 rare pLoF variants is shown.

(C) Details for the component variants are shown in a table, including their functional consequence (CSQ), a detailed protein-coding annotation (HGVSp), and the association p value and beta, as well as frequency information (AC, allele count; Hom, number of homozygotes; AN, allele number; AF, allele frequency). Each component pLoF variant in scrib has a positive beta value, and in aggregate, these variants show an association at p = 6 × 10$^{-15}$ (A).

(D) A Manhattan plot of a previous GWAS[29] of FA averaged across brain regions (top), body of corpus callosum (middle), and splenium of corpus callosum (bottom). Horizontal dashed line indicates a GWAS genome-wide significance threshold (5 × 10$^{-8}$), and vertical line indicates the location of SCRIB.

well as genes with previously established hits in the GWAS catalog and FDA-approved drug targets, show an increased phenotypic impact of pLoFs and missense variants.

**Biological insights from rare-variant association results**

The biological information encapsulated in this dataset is extremely high dimensional, and we release the full dataset of results for the benefit of the community. Here, we highlight a set of known and putative associations as examples of the power of this dataset. First, we recapitulate many known associations from previous studies, including associations between *PCSK9* and LDL cholesterol (pLoF burden p = 3.5 × 10$^{-132}$), *COL1A1* and bone density (pLoF burden p = 2 × 10$^{-9}$),[26] *KLF1* and several red blood cell traits (pLoF burden < 2 × 10$^{-12}$),[27] and *LRP5* (Wnt coreceptor) and bone density and osteoporosis phenotypes (pLoF burden < 5 × 10$^{-7}$).[28]

We highlight biological signals identified in the exome dataset, enabled by the Genebass browser. In particular, we find an association between pLoF of *SCRIB* and white matter integrity of tapetum (Figure 5). Notably, this association is not significant

at any single pLoF variant, but when aggregated into a SKAT-O or burden group test, the overall ablation of the transcript is associated at a p value of 6 × 10$^{-15}$ (Figure 5A). This provides additional context to a signal observed in a recent GWAS of white matter integrity[29] averaged across regions of the brain, as well as in the body of corpus callosum (Figure 5B). To our knowledge, this gene has not been associated in previous genome-wide association studies, although it is a constrained gene (probability of loss-of-function intolerance; pLI = 0.93) that shows evidence for neural tube defects in mice[30] with ultra-rare occurrences in humans.[31]

**DISCUSSION**

We have generated rare-variant association analysis summary statistics for 4,529 phenotypes and made these data available to the public via bulk data downloads as well as a public-facing browser (https://genebass.org). We explore aspects of this resource relating natural selection, allele frequency, and genetic discovery, and we highlight an association between *SCRIB* and a

brain imaging trait. Future work will be needed to fully assess the contribution of rare variants to the heritability of common diseases, as well as the extent and role of pleiotropy among rare variants.

### Limitations of the study

There are a number of limitations to our analysis. Although we performed extensive QC to improve the reliability of these results, the low frequency of pLoF variants and low prevalence of diseases results in instability and low power for many of the associations, even at the scale of ~400,000 individuals. Thus, we urge caution in interpreting association results, particularly for the rarest binary traits (prevalence $<10^{-4}$) and ultra-rare variants (frequency $<10^{-4}$), as the behavior of the association tests when the counts are small (the asymptotic properties) may not be met. For the rarest outcomes, increasing the number of cases is essential to properly evaluate the impact of rare coding variation across genes. Alternatively, other statistical methods such as Firth regression may be better suited to such traits. For pLoF variants, the median cumulative allele frequency across genes is approximately $1.5 \times 10^{-4}$, suggesting that group tests at current sample sizes are only powered to detect individual gene effects for quantitative traits that capture at least 0.02% of variance, as well as diseases and traits that have a high prevalence (well above 10%; Figure S10). These considerations are underscored by the apparently poor asymptotic properties of the mixed-model tests for rarer binary traits, as the lambda GC for these tests decreases precipitously (Figure S9). Nonetheless, global biological trends are apparent, such as the relative ordering of functional impact (pLoF > missense > synonymous; Figure 3), highlighting that the ability to accurately annotate variants with the functional consequences on a gene is critical to powering discovery in rare-variant analysis. Further, measures of natural selection at the gene level continue to highlight that certain classes of genes, such as LoF-intolerant genes, are clearly enriched for phenotypic associations.

Finally, these association analyses were only performed for individuals of European ancestry, the largest group in the dataset. Notably, these analyses only interrogate a slice of human genetic diversity, and expanding to additional ancestries has been shown to increase power and resolution for genetic discovery;[32–34] however, as the sample sizes of non-European individuals in the UK Biobank are very limited, these analyses would be underpowered for most binary traits including many disease outcomes. Concentrated efforts in building large biobanks with diverse participants will be required to overcome these limitations and provide more insight into the contribution of rare variants to common disease etiology.

### STAR★METHODS

Detailed methods are provided in the online version of this paper and include the following:

- KEY RESOURCES TABLE
- RESOURCE AVAILABILITY
  - ○ Lead contact
  - ○ Materials availability

  - ○ Data and code availability
- METHOD DETAILS
  - ○ Data processing
  - ○ Joint calling
  - ○ Phenotype data processing
  - ○ Genotype, sample, and variant quality-control summary
  - ○ Concordance between arrays and exomes
  - ○ Interval QC
  - ○ Sample QC
  - ○ Variant QC
  - ○ Genotype QC
  - ○ Annotations
  - ○ Scaling association testing using Hail Batch
  - ○ Association testing and quality control
  - ○ Data browser
- QUANTIFICATION AND STATISTICAL ANALYSIS
- ADDITIONAL RESOURCES

#### SUPPLEMENTAL INFORMATION

#### ACKNOWLEDGMENTS

We thank Danielle Ciofani and Cathy Marshall for their efforts in launching this project. We thank the participants and leadership of the UK Biobank: this work was done under UK Biobank applications 26041 and 48511.

#### AUTHOR CONTRIBUTIONS

Methodology, K.J.K., K.R.C., J.K.G., W.Z., and B.M.N.; software, K.J.K., K.R.C., J.K.G., W.L., J.I.G., D.K., C.V., T.P., and C.S.; validation, K.J.K. and W.L.; formal analysis, K.J.K., K.R.C., and W.L.; resources, G.T., B.M.R.-G., E.A.T., H.I.K., X.Z., F.R., S.E., A.J.G., M.R., J.W., H.J., D.S., P.G.B., X.C., X.H., J.I.G., D.K., C.V., T.P., C.S., and B.M.N.; data curation, K.R.C., G.T., D.S.P., C.C., D.P.H., C.F., C.T., and L.D.G.; writing – original draft, K.J.K, M.S., K.R.C., W.L., and B.M.N.; writing – review & editing, K.J.K., M.S., K.R.C., B.M.R.-G., X.Z., F.R., P.G.B., H.L.R., M.J.D., and B.M.N.; visualization, M.S., N.A.W., K.N., and H.N; project administration, C.M. and N.G.; supervision, D.G.M., H.L.R., A.A.P., M.J.D., J.W.D., H.R., M.R.M., and B.M.N. For a list of Biobank team members, please see the Supplemental information.

#### DECLARATION OF INTERESTS

K.J.K. is a consultant for Vor Biopharma. B.M.R.-G., X.Z., F.R., S.E., A.J.G., M.R., J.W., H.J., and J.W.D. are employees of AbbVie, Inc. M.R. is an employee of and owns stock in AbbVie, Inc. E.A.T., D.S., P.G.B., and H.R. are employees of Biogen and hold stocks/stock options in Biogen. H.I.K., X.C., X.H., and M.R.M. are employees of Pfizer. D.K. holds stock in the private company TriNetX, LLC. D.S.P. was an employee of Genomics plc. All the analyses reported in this paper were performed as part of D.S.P.'s previous employment at the Massachusetts General Hospital and Broad Institute. N.A.W. owns stock in Pfizer. L.D.G. receives funding from Intel and Illumina. D.G.M. is a founder with equity of Goldfinch Bio and serves as a paid advisor to GSK, Variant Bio, Insitro, and Foresite Labs. H.L.R. is a member of the scientific advisory board at Genome Medical. A.A.P. is a Venture Partner at GV. He has received consulting fees from Novartis and receives funding from Bayer, IBM, Microsoft, Alphabet, Intel, GSK, Pfizer, and Illumina. M.J.D. is a founder of Maze Therapeutics. B.M.N. is a member of the scientific advisory board at Deep Genomics and RBNC Therapeutics, a member of the scientific advisory committee at Milken, and a consultant for Camp4 Therapeutics, Merck, and Biogen.

## Resource

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

## Resource

CellPress

# STAR★METHODS

## KEY RESOURCES TABLE

| REAGENT or RESOURCE | SOURCE | IDENTIFIER |
|---|---|---|
| **Deposited data** | | |
| UK Biobank Exome data | UK Biobank | https://www.ukbiobank.ac.uk/ |
| Association results | This paper | https://genebass.org |
| Interval list | 41 | https://doi.org/10.5281/ZENODO.6784346 |
| **Software and algorithms** | | |
| Hail | 37 | https://doi.org/10.5281/zenodo.4504325, https://hail.is |
| PHESANT | 39 | https://doi.org/10.5281/zenodo.6795217, https://github.com/astheeggeggs/PHESANT/tree/v0.2.1 |
| SAIGE-GENE | 16 | https://www.nature.com/articles/s41588-020-0621-6 |
| Gene lists | 38 | https://doi.org/10.5281/ZENODO.6724345, https://github.com/macarthur-lab/gene_lists |
| Main pipeline and analysis code | 35 | https://doi.org/10.5281/zenodo.6726324, https://github.com/nealelab/ukb_exomes |
| Supporting code | 36 | https://doi.org/10.5281/zenodo.6726336, https://github.com/nealelab/ukb_common |

## RESOURCE AVAILABILITY

### Lead contact
Further information and requests for resources and reagents should be directed to and will be fulfilled by the lead contact, Konrad Karczewski: konradk@broadinstitute.org.

### Materials availability
This study did not generate new unique reagents.

### Data and code availability
All processed exome data has been returned to UK Biobank and will be available through their access management systems. All resulting summary statistic datasets are publicly available as bulk downloads and in a browser interface at https://genebass.org. All original code has been deposited to Zenodo with DOIs as below and is publicly available as of the date of publication. Links are listed in the key resources table.

Main pipeline and analysis code (ukb_exomes package): https://doi.org/10.5281/zenodo.6726324.

Supporting code (ukb_common package): https://doi.org/10.5281/zenodo.6726336.

PHESANT package for phenotype processing: https://doi.org/10.5281/zenodo.6795217.

Hail library for downstream analysis: https://doi.org/10.5281/zenodo.4504325.

Interval list for coverage assessment: https://doi.org/10.5281/ZENODO.6784346.

Gene lists for analysis (Figure 4): https://doi.org/10.5281/ZENODO.6724345.

We make the full dataset available in a browser framework (described below), as well as Hail formats, hosted on the Google Cloud Platform. We provide one MatrixTable for each of the single-variant and gene-based tests, as well as Hail Tables with filtering criteria for variants, genes, and phenotypes.

All code to reproduce the analyses herein is available on github: https://github.com/nealelab/ukb_exomes (https://doi.org/10.5281/zenodo.6726324)[35] and https://github.com/nealelab/ukb_common (https://doi.org/10.5281/zenodo.6726336).[36] All quality control and data analysis tasks were performed using Hail (versions between 0.2.49 and 0.2.62; https://doi.org/10.5281/zenodo.4504325).[37] Gene lists were downloaded from https://github.com/macarthur-lab/gene_lists (https://doi.org/10.5281/ZENODO.6724345).[38] Unless otherwise indicated, summarized analyses and plotting were performed in R 4.0.2, using tidyverse.

Any additional information required to reanalyze the data reported in this paper is available from the lead contact upon request.

## METHOD DETAILS

### Data processing

We re-processed CRAM files according to the GATK Best Practices, briefly, aligning reads using BWA-MEM 0.7.15.r1140 and processing reads using Picard and GATK. For each sample, variants were called using the Genome Analysis Toolkit (GATK) 4.0.10.1 with local realignment by HaplotypeCaller in gVCF mode, such that every position in the genome is assigned likelihoods for discovered variants or for the reference. We then post-processed gVCF outputs, further compressing "blocks" of homozygous reference calls to seven GQ bins: 0, 10, 20, 30, 40, 50, and 60. All analyses were performed on Google Cloud Platform.

### Joint calling

Many technologies used to represent, store, and compute on genomic data do not scale efficiently to datasets of hundreds of thousands of samples with whole exome or genome sequence data. The Hail project addresses these scaling limitations by developing new technology to enable representation and analysis of cohorts with hundreds of thousands of samples. The project VCF (pVCF) representation scales poorly, and one key innovation that allowed efficient analysis of such a large callset is the Scalable Variant Call Representation (SVCR), which takes advantage of reference block compression to guarantee strictly linear asymptotic scaling with the number of stored samples. The SVCR is a conceptual representation for the variant-level data on a sequenced cohort. In the Hail library, we have developed an implementation of an efficient hierarchical merge algorithm to create a dataset in Hail conforming to the SVCR representation from single-sample gVCFs, and a rudimentary set of functionality for working with this object as an analysis target.

#### *Representation*

We will briefly describe the representation of the SVCR and where it differs from a pVCF. First, the SVCR contains a row for each locus defined in any input gVCF. While pVCF only includes a row for each variant site, the SVCR contains rows for monomorphic sites where reference blocks begin. The SVCR contains more loci than an equivalent pVCF because while pVCFs only contain loci with genomic polymorphism, the SVCR contains loci where reference blocks begin but with no variation. This difference in locus cardinality shrinks with increasing sample size when rare variants are discovered at most positions in the genome. The alleles field of the SVCR at a given locus contains the set of all unique alternate alleles observed in any input gVCF for that locus (Figure S1).

An entry of the SVCR matrix corresponding to locus L and sample S is dense (defined) if the gVCF for sample S has a line with locus L, and is sparse (missing) otherwise. If defined, the entry includes a light transformation of all information from the gVCF for sample S at locus L, with the two exceptions of the locus information (chromosome and position) and the alleles (reference and alternates). The chromosome and position are necessarily the same as those constituting L, and instead of including alleles as strings, the SVCR has an additional field, LA, or "local alleles", which is a list of integers. This list contains the indices in the list of all discovered alleles at the current locus corresponding to alleles that were discovered in the GVCF for sample S. The final transformation is a rename of GT and all fields that are "R-" or "A-" numbered, such as AD and PL, to LGT, LAD, LPL, etc. This rename reflects the change in meaning – the alleles referred to by these fields are the ones defined in LA, rather than the full list of discovered alleles in the SVCR row.

It is important to note that the SVCR representation is a lossless transformation of gVCFs – the original gVCFs can be recreated by reversing the transformations defined above.

#### *Merge algorithm*

We have proposed the SVCR as an efficient conceptual representation for efficiently storing cohort-level data sequencing data. We now describe a hierarchical merge algorithm that lends itself to scalable construction of an SVCR.

First, each input gVCF is converted to a single-sample SVCR. This conversion involves trivial reorganizations and renames of fields (GT to LGT, as described above). The result is an SVCR with the same number of rows as the gVCF and one column, where every entry is defined.

Second, N SVCR are merged together into a new SVCR. First, an outer join on loci is executed for the N SVCRs, producing some number of intermediate rows with a locus (the join key), the alleles for each input SVCR (which may be missing if an SVCR did not contain that locus), and the entries associated with that locus for each SVCR. The merged alleles are computed by taking the set of all unique alleles from all input SVCR matrices, and sorting according to a deterministic string ordering function. The merged entries are computed by taking the full set of entries across join inputs, and updating one field: the values of each LA (local alleles) field must be recomputed to refer to the new, merged alternate alleles.

The Hail framework features a concrete implementation of this hierarchical merge algorithm, repeating the second step in rounds until a single SVCR remains. In practice, this algorithm defaults to a value of 100 for the branch factor parameter, N, which performs well empirically and allows for the creation of a million-sample dataset in three rounds of merging.

#### *Dense matrix analysis*

The significant scaling advantages of the SVCR representation are not without cost. The sparsity prevents random access to reference block information at any specific variant, because most of the reference blocks spanning that variant will have been defined at earlier loci. Analyses that require access to reference block metadata (such as GQ or DP) can be implemented following a "densification" pass that carries forward an array of genotype values, one for each sample, corresponding to the most recent reference block entry on the same chromosome. The reference blocks in this array can be used to fill sparse entries for each locus in the SVCR (as long as a candidate reference block spans the locus of interest). The necessary dense analysis target can thus be realized at low cost

on the fly during analysis pipelines, then discarded. It is not necessary to save the dense matrix to durable storage, which would incur prohibitive cost.

### Phenotype data processing

To automate the curation and harmonization of the large collection of variable scalings, encodings, and follow-up responses in a coherent manner, we created a modified version of the PHEnome Scan ANalysis Tool (PHESANT), available at https://github.com/astheeggeggs/PHESANT (https://doi.org/10.5281/zenodo.6795217).[39] Unlike the original implementation,[40] our version does not perform association analyses, but simply generates a collection of re-coded phenotypes. An outline of the PHESANT pipeline and our chosen filters are displayed in Figure S2. We manually curated the collection of phenotypes for study, which we filter to as part of the pipeline. Re-codings of variables, and inherent orderings of ordinal categorical variables, are defined in the data-coding file, which is available in our GitHub repository.

In addition to the inverse-rank normalization applied to the collection of continuous phenotypes, we also retain the raw version of the continuous phenotype, with no transformation applied to the data (though re-codings of the data to guard against spurious results are retained). Following all of these alterations and additions, we run this modified version of PHESANT on the phenotypes in our UK Biobank application using a 200Gb RAM virtual machine on the Google Cloud Platform.

Upon applying the PHESANT pipeline to our selected collection of phenotypes, a small subset of categorical variables remain that should be sex-specific but are not excluded from the both-sexes collection of phenotypes. We manually identified this collection of sex-specific phenotypes and removed them from the both-sex phenotypes before subsequent analysis.

We loaded all phenotype data into a Hail MatrixTable using a custom processing script (https://github.com/Nealelab/ukb_exomes/blob/master/hail/load_phenotype_data.py [https://doi.org/10.5281/zenodo.6726324][35]). Briefly, we parsed the PHESANT output, extracted ICD codes from the "first occurrence" data (which were run as binarized outcomes), parsed some custom phenotypes and covariates, and combined these into a phenotype MatrixTable. The MatrixTable is keyed by trait type (continuous, categorical, and ICD code), phenocode, sex, coding (for categorical traits), and modifier (raw vs inverse-rank normalized for continuous traits). For this supplement, we use the descriptors "categorical" and "binary" interchangeably when describing phenotypes.

### Genotype, sample, and variant quality-control summary

We performed quality control (QC) in a similar fashion to the approach used for the Genome Aggregation Database (gnomAD).[15] Notably, however, we included a number of additional metrics, including concordance between arrays and exomes and interval QC.

### Concordance between arrays and exomes

We confirmed that all 453,644 samples with available array data had a high proportion of variant concordance between their exome and array data. We filtered the UK Biobank array data to autosomal regions and lifted the data to genome build GRCh38 prior to examining concordance. The minimum proportion concordance of non-reference sample genotypes was 0.96 (mean 0.995).

### Interval QC

We performed quality controls on intervals targeted by the exome capture before applying sample hard filters. In this interval QC, we investigated how much padding to add around the UK Biobank capture intervals and whether to check coverage across standard Broad exome calling intervals (available on Google Cloud Platform at gs://gcp-public-data–broad-references/hg38/v0/exome_calling_regions.v1.interval_list [https://doi.org/10.5281/ZENODO.6784346][41]) as well. We determined that padding more than 50 base pairs into the intron added too much noise and reduced sample call rate and coverage significantly. We also discovered that adding calling intervals unique to the Broad's set of exome targets also reduced call rate and coverage. Therefore, we decided to keep variants only within the 50 base pair padded UK Biobank intervals. For sample QC, we also decided to filter to intervals where 85% of samples had a mean coverage of at least 20X (Figure S3).

### Sample QC
#### Sex imputation

We used Hail's *impute_sex* method to infer sex using common (allele frequency >0.1%), non-pseudoautosomal, biallelic single nucleotide variants (SNVs) on chromosome X. We then computed non-pseudoautosomal chromosome X and chromosome Y coverage for each sample and normalized these values using sample-specific coverage across chromosome 20. We then checked the distribution of chromosome X and Y ploidies for XX and XY karyotypes to determine each karyotype's ploidy cutoffs (upper cutoff for single X: 1.55, lower cutoff for double X: 1.6, upper cutoff for double X: 2.15, lower cutoff for triple X: 2.2, lower cutoff for single Y: 0.06, upper cutoff for single Y: 1.3, lower cutoff for double Y: 1.4). The adjusted ploidy cutoffs helped us add additional granularity to the inferred sex, differentiating between X0, XX, XXX, XY, XXY, XYY, and XXYY karyotypes (Figure S4).

#### Hard filters

We applied three hard filters post-interval QC to the samples: sex imputation filters (removing any samples that were not inferred as XX or XY), a call rate filter (cutoff: 0.99), and a coverage filter (mean coverage cutoff 20X). We excluded hard-filtered samples from platform PCA, relatedness inference, ancestry imputation, and outlier detection so that these low quality samples would not influence our downstream results.

### Platform inference

Although we do not expect there to be noticeable technical artifacts given that the samples were run on the same platform, we ran a principal component analysis (PCA) on the per-individual per-interval call rate matrix, as previously described,[15] to make sure there were no significant clusters. Our platform PCA picked up a few clusters and showed some differences between the different sample batches on PCs three and 4 (Figure S5). After some investigation, we discovered that the PC4 separated samples due to a common CNV (chr12:9531541-9531809). As PC3 showed some separation by batch, we decided to use batch status as a proxy for platform in further sample QC and a covariate for association analysis.

### Relatedness inference

We used Hail's *pc_relate* method to infer relatedness on SNVs that are autosomal, bi-allelic, common (allele frequency >0.1%), high call rate (>99%), and LD-pruned with a cutoff of $r^2 = 0.1$. We then used the *maximal_independent_set* method in Hail to keep the largest set of unrelated samples (samples without second degree or closer relationships; kinship >0.1), prioritizing samples with greater mean depth. The related samples that were not in the maximal independent set were flagged.

The majority of the samples removed during sample QC were samples inferred to have a second-degree or closer relationship with other samples in the dataset. This finding is not unexpected, as previous studies have shown that the UK Biobank data contains a large number (approximately 30% at third-degree or greater) of related samples.[7]

### Ancestry imputation

We used a hybrid method to infer ancestry for sample QC. We projected the UK Biobank data onto the gnomAD population principal components (PCs) and then used a random forest classifier trained on gnomAD ancestry labels to assign ancestry to the UK Biobank samples. We observed that many samples labeled as African using this method were flagged as outliers by our downstream population-stratified outlier detection method. This seemed to be due to the fact that one cluster of samples labeled as African appeared highly admixed. To account for this, we ran a PCA on the UK Biobank samples and applied a clustering method (HDBSCAN). We found that this clustering method split the African labeled samples into additional clusters and reduced the number of samples flagged as outliers while also recapturing most of the same global population clusters observed in gnomAD.

As a result, we chose to assign ancestry using a hybrid of the projection onto gnomAD PCs and the UK Biobank specific PCA clustering: for any sample that was assigned to a cluster using the UK Biobank PCA, the sample was given that cluster as their ancestry assignment to preserve the sub-structure observed using clustering. Any sample that was not assigned to a cluster was given the label from the initial (gnomAD) PCA projection and random forest classification.

We obtained ancestry labels from the Pan-UK Biobank project (UK Biobank Return 2442) after completing sample QC, and these labels were used in all downstream analyses.

### Outlier detection

We flagged any sample falling outside 4 median absolute deviations (MADs) from the median of any of the following metrics (stratified by population and tranche as a proxy for platform), which were calculated using hail's *sample_qc* method:

- Number of deletions.
- Number of insertions.
- Number of SNVs.
- Insertion: deletion ratio.
- Transition: transversion (TiTv) ratio.
- Heterozygous: homozygous ratio.

The final counts of samples is shown in Table S1.

### Variant QC

Variant filtering consisted of a combination of a random forest (RF) classifier and hard filters. We used the following training sets as true positives for training the random forest model:

- Omni – SNVs present on the Omni 2.5 genotyping array and found in 1000 Genomes data.
- Mills – Indels present in the Mills and Devine data.[42]
- Transmitted singletons – Variants found in only two individuals, which were a parent-offspring pair.
- Sibling singletons – Variants found in only two individuals, which were a sibling pair.
- Common (AF > 0.1%) and concordant (>90% non-reference concordance) array variants.

For the false positive training set in the random forest model, we used variants that fail traditional GATK hard filters: QUAL by depth (QD) < 2, strand bias estimated using Fisher's exact test (FS) > 60, or root mean square mapping quality (MQ) < 30. We balanced the number of variants in the true positive and false positive training sets by randomly downsampling the false positive training set to the same number of variants found in the true positive training set. RF training was performed on only variants that fall within intervals that pass interval QC (described above; intervals where >85% of samples have a mean coverage >20X).

We used the following allele and site annotations as features in the random forest model (RF feature importance shown in Figure S6):

- Allele type – SNV, indel.
- Number of alleles – Total number of alleles present at the site.
- Variant type – SNV, indel, multi-SNV, multi-indel, mixed.
- Mixed site – Whether more than one allele type is present at the site.
- Spanning deletion – Whether there is a spanning deletion (STAR_AC > 0) that overlaps the site.
- Quality by depth – Sum of the non-reference genotype likelihoods divided by the sum of the depth in all carriers of that allele.
- Read position RankSum – Rank-Sum Test for relative positioning of reference versus alternate alleles within reads.
- Mapping quality RankSum – Rank-Sum Test for mapping qualities of reference versus alternate reads.
- Strand bias odds ratio – Symmetric Odds Ratio test of 2x2 contingency table to detect strand bias.
- Max probability of allele balance – Highest p-value for sampling the observed allele balance under a binomial model with p = 0.5 (maximum across heterozygotes).

RF probability cutoffs for calling a variant PASS were chosen to maximize sensitivity and specificity based on criteria such as the number of *de novo* mutations found in the 224 trios in the dataset and precision-recall curves (Figure S6B) in two truth samples present in our data (NA12878 and a pseudo-diploid sample (syndip); syndip was sequenced at Broad, not with the UK Biobank cohort). Final thresholds were RF true positive probability of 0.061 (approximately 90% of SNVs in well-covered intervals) for single nucleotide variants and 0.064 (approximately 75% of indels in well-covered intervals) for indels. Finally, we also excluded variants with two hard filters:

- Excess heterozygotes defined by an inbreeding coefficient < −0.3.
- Variants where no sample had a high quality genotype (see Genotype QC below).

### Genotype QC
We filtered genotypes based on the previously defined "adj" criteria, with a modification for haploid calls on chrX and chrY for XY individuals. Specifically, we filtered to genotypes where depth $\geq$ 10 (5 for haploid calls), genotype quality $\geq$ 20, and minor allele balance >0.2 for all alternate alleles for heterozygous genotypes.

### Annotations
Variants were annotated using VEP v95 as implemented in Hail using the default parameters for GRCh38 (including LOFTEE[15]).
For downstream analyses, variants were grouped by Ensembl Gene ID and functional impact as follows:

- pLoF: High-confidence LoF variants (as indicated by LOFTEE), including stop-gained, essential splice, and frameshift variants, filtered according to a set of first principles as described at https://github.com/konradjk/loftee.[15]
- missense|LC: Missense variants are grouped with in-frame insertions and deletions, as well as low-confidence LoF variants (filtered out by LOFTEE). The latter have a frequency spectrum consistent with missense variation,[15] and affect a set of amino acids in a similar fashion (e.g. a frameshift in the final exon).
- synonymous: All synonymous variants in the gene.

### Scaling association testing using Hail Batch
To perform the association analysis using SAIGE-GENE, we developed a new scientific compute scheduler, Hail Batch. Hail Batch is a cloud-based, serverless, multi-tenant platform as a service (PaaS). To use Hail Batch, users construct a computational graph of jobs to be executed, called a batch, using a Python client library (or manually). The batch is then submitted to Hail Batch via a REST API. The Hail Batch scheduler both manages pools of worker virtual machines (VMs) on which to schedule user jobs and schedules jobs on those workers. Hail Batch includes a Web UI for monitoring batches and viewing individual job logs. The documentation can be found here: https://hail.is/docs/batch/index.html (https://doi.org/10.5281/zenodo.4504325).[37] Hail Batch is fully open source and is contained in the Hail project monorepo which can be found here: https://github.com/hail-is/hail/tree/main/batch (https://doi.org/10.5281/zenodo.4504325).[37]

*Motivation*
We built Hail Batch because we wanted a serverless solution with zero operational overhead for users. Based on our experience building Hail Query on Apache Spark, even with cloud platform managed services for running Spark like Google Dataproc, configuring, provisioning and managing compute clusters is a significant operational burden each user must bear (and scales with the number of users). In addition, by operating a multi-tenant compute cluster, we increase the utilization of resources amongst all users. The benefits are especially pronounced when users are working on iterative analyses (as opposed to batch processing) where they might need time to assess a result before moving on to subsequent analyses.

Bioinformatic tools come in a variety of forms, from standalone binaries and command line tools like GATK and SAIGE, to Python and R packages, to cloud-native analytics tools like Hail Query. We wanted a scheduling infrastructure that would support building data processing pipelines across all these tool modalities. Hail Batch currently supports executing containerized command line tools and serialized Python functions, as well as a Hail Batch based backend for Hail Query which executes JVM bytecode.

Finally, we wanted to support low scheduling overhead at a large scale so that users could decompose pipelines based on natural biological or data considerations rather than computational constraints. For example, running millions of relatively fast statistical tests

(seconds or more) for permutation testing requires small scheduling overhead to be effective. The association analysis described here naturally decomposed into a job per megabase per phenotype, for a total of approximately 18 million jobs.

We follow the principle that the systems we build today should themselves become building blocks of the systems we build tomorrow just as the association pipelines described herein build on Hail Batch. Therefore, we wanted a system that had a native program programmatic interface so it could be used by other systems. For example, the Hail team's continuous integration (CI) system runs tests and deployments using Hail Batch. A system like Hail Batch can also serve as the underlying execution engine for an incremental joint caller service.

### User interface

Hail Batch provides a Python library to construct and execute batches. A simple example is shown in Figure S7. Rather than focus on the Python interface, which is described in detail in the documentation, we will focus on the main conceptual pieces of a Hail Batch computational graph.

A batch is the unit of submitted work. A batch primarily consists of (1) user-defined attributes, and (2) an ordered list of N jobs. The user-defined attributes are for searching and identifying batches. Jobs are the individual units of scheduling and can depend on previous jobs to form a directed acyclic graph (DAG) representing the full computational pipeline. The main parts of the job description are: (1) user-defined attributes, (2) dependencies, (3) inputs and outputs, (4) compute resources, and (5) executor configuration. Jobs do not run until all of their dependencies have completed. There are mechanisms for cancellation and controlling whether jobs run if their dependencies failed or were canceled. Compute resources (CPU, memory, disk and machine type) describe resources needed to schedule the job, which will be provisioned by the autoscaler as needed. We refer to the reader to the documentation for more details.

During the submission and execution of a batch, jobs pass through the following states: pending (dependencies are not complete), ready (dependencies are complete and the job is ready to be scheduled), creating (the job is provisioning resources for custom resource configurations), running (the job has been scheduled), and the terminal states: error (the executor failed to run the job), canceled, failed (the job execution failed), or successful. The REST API and Web UI allow users to monitor running and historical batches and view individual job logs.

There are currently two backends for executing a batch: (1) a local backend that runs jobs locally and (2) a service backend that submits jobs to the Hail Batch service. Currently, the Hail Batch service supports the Google Cloud Platform.

### Implementation

Hail Batch is written in Python and makes extensive use of asyncio. Its deployment consists of three parts: (1) the front-end **batch** service, (2) **batch-driver** service, and (3) a MySQL database instance. The **batch** service serves the Web UI and user-facing REST API queries. It is stateless and autoscales based on incoming traffic. The **batch-driver** runs the job scheduler, the autoscaler which provisions worker VMs, and the admin Web interface. Job and batch configuration state is stored in MySQL and job logs are stored in Google Storage. Hail Batch relies on a Hail PaaS **auth** service for session ID based authentication. The Hail PaaS services are deployed in Kubernetes and the autoscaler uses the Google Compute Engine (GCE) directly to provision workers. Hail Batch deployment is controlled by the **ci** (continuous integration) service. We also maintain Terra-form scripts for bootstrap and disaster recovery of Hail PaaS installations.

The autoscaler in the driver service is organized in terms of instance pools. There are two types of instance pools, shared and private. There are three shared pools for the three main GCP machine types. Shared pools autoscale based on the pool size and the number of ready jobs that run in that pool. Batch pools can scale down to zero, but we also support a minimum size so small jobs can be dispatched immediately without waiting to spin up resources. Shared pool workers schedule jobs across all users and resource utilization is higher because instance startup and shutdown is generally amortized over many jobs. By default, preemptible n1-standard-16 machines are used, but jobs that use special machine types or require non-preemptible instances can run in a private pool which provisions a VM per job.

In addition to autoscaling, the Hail Batch scheduler implements a fair share algorithm across users to provide a responsive experience. Jobs are scheduled per user, in reverse order of the amount of resources already allocated. If there are not enough ready jobs to saturate a user's allocation, the unused allocation is made available to the remaining users. This means new jobs submitted by a user to an active cluster will scale up quickly to the user's share and all users enjoy a more responsive experience when the system is actively used.

We make a few remarks on the scheduler performance. The scheduler is able to schedule $\sim$80 jobs/s. The maximum cluster size the scheduler can support is a function of the expected job length. The maximum cluster size is (average job length in second) * 80. Therefore, for the 15–30m SAIGE jobs used in the association analysis here, the scheduler is theoretically (and was practically) able to saturate a $\sim$100K core pool of workers.

Hail Batch currently supports two executors on worker VMs: Docker and the Java Virtual Machine (JVM). The JVM executor is used by the Hail Query service. The Docker executor includes the details necessary to run a docker container: image, environment variables, command line, etc. Jobs run on the worker VMs in three steps: input, main and output. The input and output steps are responsible for copying between object storage and the local filesystem. For copying, we developed a pluggable Python asyncio filesystem abstraction and high-performance parallel copy management engine which supports local files, Google Cloud Storage, S3 and HTTP(S) (read-only). The main step executes the user's code.

## Association testing and quality control
### Association testing framework
We performed association testing on the quality-controlled genotype data using SAIGE-GENE,[43] following the recommendations by the authors. We computed a genetic relatedness matrix (GRM) using a dataset sampled from allele frequency categories from the genotype MatrixTable considering only autosomal variants with a minimum call rate of 95%, including approximately 2,000 variants from each of Allele Count (AC) 1-5, AC 6-10, and AC 11-20; and approximately 10,000 variants from each of: AC 20-AF 0.1%, AF 0.1-1%, AF 1-10%, AF > 10%. These variants were LD-pruned to $r^2 = 0.1$, and exported into PLINK format. We computed a sparse GRM using step0 of SAIGE-GENE using the default parameters, with 2,000 markers used for the kinship matrix, and a relatedness cutoff of 0.125. We further created a "gene map" file for each megabase, which included information about the variants to be analyzed together in each group test. We included 3 groups: pLoF variants, including only those annotated as high-confidence by LOFTEE; missense-like variants, including missense variants and variants annotated as low-confidence by LOFTEE; and synonymous variants. The script for pre-processing is available at https://github.com/Nealelab/ukb_exomes/blob/master/hail/pre_process_saige_data.py (https://doi.org/10.5281/zenodo.6726324).[35]

The remainder of the process was parallelized using Hail Batch (Figure S7). For each megabase of the genome, we exported a BGEN from the genotype MatrixTable with all variants that lie in genes that have a starting coordinate within that megabase. For each phenotype, we exported a flat file from the phenotype MatrixTable with the covariates used for analysis: age, sex, age,[2] and 20 principal components, as well as interaction terms of age * sex and age[2] * sex. The phenotype data were combined with the sparse GRM computed above to fit a null model (without genotypes) using step1 of SAIGE-GENE with the default parameters and the covariates described above. Finally, we ran the genotype tests using the BGEN from each megabase, and the null model from each phenotype using step2 of SAIGE-GENE with the default parameters, plus maxMAFforGroupTest = 0.01, maxMAF = 0.5, LOCO = FALSE, and IsSingleVarinGroupTest = TRUE. The results across all megabases were loaded into two Hail Tables for each phenotype, for the group tests and single variant tests. The pipeline is available at https://github.com/Nealelab/ukb_exomes/blob/master/saige_exomes.py (https://doi.org/10.5281/zenodo.6726324)[35] with helper scripts that can be found at https://github.com/Nealelab/ukb_common/blob/master/utils/saige_pipeline.py (https://doi.org/10.5281/zenodo.6726336).[36]

We combined the phenotype-level Hail Tables into a Hail MatrixTable using a hierarchical merge, along with phenotype metadata from SAIGE, resulting in one MatrixTable for the group tests and one for the single variant tests. We computed lambda GC values for each phenotype and gene (see below) using the *hl.methods.statgen._lambda_gc_agg* aggregator in Hail. These datasets are publicly released and serve the browser framework described below.

### Random phenotype analysis
To test the asymptotic properties of our tests, we simulated 314 random normally distributed phenotypes in the 300K callset of this resource with a range of heritabilities using the sparseMVN package in R,[44] using the genetic relatedness matrix generated by SAIGE. From these normal distributions, we simulated continuous phenotypes, as well as binary phenotypes with varying prevalences from $10^{-4}$ to 50%, with heritability of 1. We further generated a series of phenotypes of varying heritabilities (0.1, 0.2, 0.5) by introducing an additional noise component (rnorm) and weighting by the square root of the desired heritability. We performed association testing on these phenotypes and computed lambda GC values as above, and here, we show the qq-plots for the single-variant and group (SKAT-O and Burden) tests (Figure S8).

In order to visualize the behavior of the tests across frequency strata in the association data, we compute the cumulative allele frequency (CAF) for each gene by aggregating the allele frequencies of variants of the same annotation group within each gene. We summarize the lambda GC metric for each trait for variants with a CAF category, which is shown in Figure S9. Notably, in Figures S8 and S9, we can see increased instability of the QQ-plot and lambda values for rarer variants especially for rarer outcomes, suggesting the need for an allele frequency threshold for large-scale analyses. This is consistent with the minimum frequency and prevalence required to achieve statistical significance at this sample size (Figure S10).

To compute an effective number of tests and thus a p-value threshold for each phenotype, we performed association testing between the genotype data and each of these phenotypes. The most significant p-value for each of the simulated continuous phenotypes is an estimator for the inverse of the effective number of tests: the median of this value across all simulated continuous phenotypes was $5 \times 10^{-6}$ for SKAT-O tests, $1.3 \times 10^{-5}$ for burden tests, and $1.6 \times 10^{-7}$ for single-variant tests. Thus, for downstream analyses, we computed the experiment-wise significance threshold as 0.05 *p, or: $2.5 \times 10^{-7}$ for SKAT-O tests, $6.7 \times 10^{-7}$ for burden tests, and $8 \times 10^{-9}$ for single-variant tests.

### Expected allele count filters
Using the CAF for each gene and number of cases (n_cases) of the phenotypes, we then defined an element-level filter, the expected allele count, as CAF × n_cases (for continuous traits, this is the CAF × the number of individuals with a defined value). For each phenotype in each group test (SKAT-O, SKAT, and Burden test), we filtered the test results to genes from each of the expected AC intervals: [0, 5], (5, 50], (50, 500], (500, 5000], (5000, 50,000], and (50,000, ∞) and computed lambda GC values for each phenotype within each bin. Lambda GC values for categorical phenotypes converge to around 1 as expected AC increases (Figure S11). Lambda GC values for continuous phenotypes are not significantly affected by the change of expected AC. A smaller number of cases and lower level of expected AC results in less stable values of lambda. Because of the highly deflated pattern of lambda GC values observed in genes with expected AC from 0 to 50, we filtered out summary statistics with expected AC < 50. For the

variant-level results, we defined the expected AC as MAF × n_cases and computed lambda GC across different intervals. We observed a similar deflated pattern at low levels of expected AC and filtered out summary statistics with expected AC < 50.

### Test calibration filtering

Removing summary statistics with expected AC < 50 results in improved lambda GC values (Figure S12). Further, we filtered association statistics with standard errors (SE) of 0, as these were the result of a bug in the version of SAIGE-GENE used herein (which has since been corrected).

Finally, due to the large number of phenotypes available, we devised a metric for the calibration of individual genes, a lambda GC for each gene across phenotypes. For most genes, this metric is well-behaved for synonymous variants (95% range). However, it also marks outliers for removal, and appears to be correlated with mean sequencing coverage (Figure S13). Thus, we removed genes with coverage <20 as well as genes that have a synonymous lambda GC < 0.75.

After applying the above filters, we recompute lambda GC for each phenotype (Figure S14) and gene (Figure S15). For downstream analyses, we filter to phenotypes with lambda GC at least 0.75.

### Independent phenotypes

For large-scale analyses, we pruned to a set of relatively uncorrelated phenotypes. Using the UKB phenotype MatrixTable, we generated a pairwise correlation table using a matrix multiplication of the table and its transpose (Figure S16A), and filtered the table to phenotype pairs with correlations ($r^2$) over 0.5. We then applied the *maximal_independent_set* function in Hail to the remaining phenotype pairs with a tie-breaker function preferring phenotypes with more cases, resulting in a set of 640 related phenotypes to remove from the dataset (Figure S16B). A summary of the final QC steps is shown in Table S2.

### Comparison to known hits

We compared the significant associations with height discovered from our results with the 91 height-associated variants ($p < 2 \times 10^{-7}$) and 10 height-associated genes ($p < 5 \times 10^{-7}$) discovered in GIANT.[17] Among the 91 rare/low-frequency variants associated with adult height in GIANT, 50 of the variants were found to be associated with height at $p < 8 \times 10^{-9}$, and 78 variants were found to be associated with height at $p < 0.05$ in UK Biobank (Tables S3 and S4). Among the 10 genes associated with height in GIANT, FLNB (pLoF, missense|LC), NOX4 (missense|LC), OSGIN1 (missense|LC), and UGGT2 (pLoF) are found associated with height at $p < 2.5 \times 10^{-7}$ (SKAT-O) or $p < 6.7 \times 10^{-7}$ (Burden), and all genes were found at nominal significance ($p < 0.05$) for either missense or pLoF variants (Table S5).

We compared the effect sizes from our single-variant test results to those of GIANT. Among 1,330 ExomeChip variants with p-value $< 2 \times 10^{-7}$ in the GIANT European-ancestry meta-analysis, 496 variants are tested in the UK Biobank data. The effect sizes of this group of shared variants are consistent between the two datasets, with a slight attenuation in results from UK Biobank, consistent with a degree of winner's curse (Figure S17).

Finally, we compared associations for 7 red blood cell phenotypes discovered in our results with 20 associations ($p < 5 \times 10^{-9}$) between missense variants and red blood cell phenotypes discovered by TOPMed[18] and find that 19 out of 20 are replicated at $p < 0.05$ with 9 of these associated at $p < 8 \times 10^{-9}$ (Table S6).

### Analysis of summary statistics

We performed all downstream analyses in Hail using the single-variant and group-test MatrixTables as described above. For these analyses, we assess the proportion of genes or variants that reach our p-value threshold for at least one trait. Notably, when simply computing this metric across functional classes, seemingly paradoxically, we observe that the proportion of associations does not correlate with the expected deleteriousness of the variants (pLoF, followed by missense, followed by synonymous) for gene-based or single-variant tests (Figure S18).

However, considering the observation that functional pLoF variants tend to be rare, and thus, have lower power for genetic discovery, we performed a series of analyses incorporating the frequency of variants and found that the former observation is due to Simpson's paradox. First, when separating genes by frequency strata, we observe the expected pattern where pLoFs variant groups have the highest proportion associated within each frequency group (Figure 3B). For high CAF genes and common variants (>1%), this trend is no longer apparent, likely complicated by the presence of artifacts for common pLoF variants (Figure S19). Second, we compare gene sets by a sampling methodology to match the CAF of gene sets to a background distribution (Figure 4 and below).

### Gene-set analyses

We considered the SKAT-O association results for 470 genes previously implicated in developmental delay[25] and compared the number of associations discovered for these genes with the remaining genes in the dataset. To match the background distribution on frequency, we binned genes by their cumulative allele frequency into equal-spaced groups with widths of 0.01, and then matched genes from the remaining set to the distribution of the 470 genes according to their CAF intervals. For each of the three annotation categories (pLoF, missense|LC, and synonymous), we randomly sampled 1,000 matched sets of 470 genes from the remaining set with replacement and computed the mean number of associations and the proportion of genes with at least one association meeting our p-value threshold for each set. By comparing the distribution of the mean and proportion of the 1,000 samples with those of the 470 genes by annotation groups, we found that genes that are implicated in developmental delay are more likely to be associated through a pLoF mechanism with phenotypes in the UK Biobank ($p = 6.6 \times 10^{-4}$, OR = 6.16; Figure 4).

Similar to the developmental delay genes, we compared 3,582 (1,701 unique genes) constrained gene-annotation pairs from our dataset with the remaining unconstrained genes on their number of associations discovered from SKAT-O results. We obtained LOEUF values for the genes from gnomAD (v2.1.1) and defined constrained genes as those in the highest decile of LOEUF

(oe_lof_upper_decile = 0). We then matched the unconstrained genes to the constrained genes by CAF intervals with widths of 0.01 and randomly sampled 1,000 unconstrained gene-sets that have sample sizes and CAF distributions comparable to the constrained gene-set for each annotation category with replacement. Finally, we compared the mean number of associations and the proportion of genes with at least one association meeting our p-value threshold of the constrained set with the distribution of the 1,000 unconstrained samples. We found that constrained genes are more likely to be associated with a phenotype in UK Biobank than the unconstrained genes, for pLoF variants (p = 1.2 × 10$^{-3}$, OR = 3.14; Figure 4).

We repeat this analysis for a number of gene sets, including autosomal recessive, autosomal dominant, FDA approved drug targets, and GWAS catalog genes, as described in https://github.com/macarthur-lab/gene_lists (https://doi.org/10.5281/ZENODO.6724345).[38]

### PolyPhen2 predicted variants

We compared the proportion of variants with at least one association meeting our p-value threshold across the three PolyPhen2 prediction groups (probably damaging, possibly damaging, and benign; Figure S20). We binned variants by their minor allele frequency (MAF) into equal-spaced bins with width of 0.01. Using variants from each one of the three groups as reference, we matched the remaining two groups to the reference group by their MAF bins. Relative relationships of the proportion among the three groups are similar when using different reference groups. We then split the variants into allele frequency categories and compared the proportion among different PolyPhen2 prediction groups. We conducted a pairwise proportion test between each pair of groups for each allele frequency interval and observed a significant difference between benign and probably damaging for all three intervals, possibly and probably damaging for allele frequency interval (0.01%, 0.1%] , and benign and possibly damaging for allele frequency interval (0.01%, 0.1%] and (0.1%, 1%]. No significant group difference is observed for allele frequencies above 10%.

### ClinVar variants

We obtained pathogenicity of variants from the ClinVar table in gnomAD reference data and then defined pathogenic and likely pathogenic variants as P/LP, benign and likely benign variants as B/LB. We divided the minor allele frequency (MAF) of variants into equal-spaced bins with widths of 0.01 and then matched variants from B/LB and Uncertain significance group to the 864 P/LP variants respectively by their MAF bins. For each of the categories, we randomly sampled 1,000 sets matched to the P/LP variant subset with replacement and computed the mean number of associations and the proportion of variants with at least one association meeting our p-value threshold for each set. By comparing the distributions of the mean and proportion of the 1,000 samples with the P/LP group, we found that pathogenic variants are more likely to be associated with a phenotype in the UK Biobank (Figure 3).

### Data browser

Web-based tools like PheWeb have been highly useful in the data processing and dissemination of several recent large-scale biobank genetic studies.[45,46] PheWeb is well-suited for viewing associations from genotyping data along large genomic regions, where the signal is frequently driven by non-coding regulatory variants rather than variation in protein coding sequences. New web-based tools are needed for visualizing association studies in the context of gene-based analyses. Toward this goal, we previously extended on our gnomAD browser toolkit[15] to create a suite of portals for displaying gene analysis results from psychiatric exome association studies for schizophrenia,[47] autism,[48] bipolar disorder,[49] and epilepsy.[50] In this study, we extend our exome browser toolkit to support visualization of biobank-scale PheWAS results. We developed new layouts, navigational mechanisms, plots, and controls that enable users to visualize and compare gene and variant associations across thousands of phenotypes.

### Navigation and workflow

The browser interface features a novel split-screen design for rapidly inspecting gene-based PheWAS results (Figure S21). The left hand side displays the global results index, which displays all hits for a given gene, phenotype, or variant (Figure S21A). The results index displays PheWAS plots or Manhattan plots depending on which navigational button is selected in the top bar (Figure S21B). The results pane can be condensed, expanded, or hidden entirely by clicking the presets buttons or by dragging the central dotted line left or right. Clicking on one of the arrow buttons in the phenotype table will update the right hand side of the PAGE with a detailed view of the selected gene-phenotype relationship (Figure S21C). The status bar will update accordingly to reflect the gene, phenotype, variant, or burden dataset that is currently selected (Figure S21D). Most data are served by a Hail backend, with significant associations cached for speed. Pages load quickly (<1 s) if the phenotype-gene or phenotype-variant association p-value pair is below the cache threshold (10$^{-4}$ for genes, 10$^{-6}$ for variants) or ∼4 s if above the cache threshold. Partitioning the PAGE in this way allows users to quickly inspect many associations without losing a sense of context, and either half can be easily hidden to create more screen room for information of interest.

### Exploring associations by gene

When the "gene PheWAS" results pane is active, the results index displays all phenotypes associated with a particular gene in PheWAS plot and tabular formats (Figure S21C). The phenotype control panel (Figure S21E) enables users to specify which of the three burden tests (Burden, SKAT, SKAT-O) or mutational class (pLoF, missense, synonymous) test statistics to display. Phenotypes can be filtered by keywords such as phenotype description or trait type (continuous, categorical, or ICD10). The results can also be filtered by p-value or beta using minimum and/or maximum thresholds. Note that for genes, the beta statistic is always derived from the burden test (SKAT and SKAT-O do not produce beta statistics). The PheWAS plot is colored and grouped by UK biobank showcase category; the category control section can be used to traverse the showcase tree and filter the phenotype list to those belonging

to specific categories. The PheWAS plot can be configured to show p-values on either log or double log scales. Users can expand the plot to focus on p-values only, betas-only, or view p-values and betas simultaneously.

The gene burden statistics table summarizes burden results across all mutational classes and tests (Figure S21F). The gene plot displays single variants mapped to genomic coordinates along the gene exons. Variant -$\log_{10}$p values are shown on the Y axis (Figure S21G). The plot transitions from a single log scale to a double log scale $\frac{2}{3}$ along the plot height to prevent variants with extremely low p-values from dominating the plot, allowing users to focus on novel rare variant associations near the significance threshold. Variants are depicted as circles, with the circle radii log-scaled by allele frequency in the non-Finnish European population. By default, variants are colored by their most severe VEP consequence across transcripts. If the selected phenotype is categorical, two additional case/control variant tracks display variant positions with radii log-scaled to allele frequencies in cases and controls, respectively. If the selected phenotype is continuous, variant radii will be log-scaled by allele frequency among individuals measured for the trait.

The single variant analysis control panel is used to configure data displayed related to single variants (Figure S21I). Checkboxes enable users to filter variants to those included in the gene burden analysis. Variants are filterable by identifier or annotation using the search box. Users can focus on particular parts of the allele frequency spectrum by dragging the allele frequency filter slider. Detailed summary statistics for all exome variants are available in a table below the plot (Figure S21H). Users can specify which columns to display using the column selection checkboxes, or they can choose one of the column group presets. Each preset will select a particular set of columns that make sense to compare side-by-side (e.g., allele counts, frequencies, population counts, and columns best suited for categorical or continuous trait types).

### Exploring associations by phenotype

When the "gene Manhattan" results pane is active, the results index displays all gene associations with a selected phenotype. The results are displayed in Manhattan plot, QQ plot, and tabular format (Figure S22). The three burden test types are displayed as columns, and the burden set (pLoF, missense|LC, or synonymous) can be selected with the "Burden set" segmented control. Clicking on a gene name will navigate to the gene PheWAS view, and clicking on the "details" arrow will update the right hand side without leaving the gene Manhattan view. When the "variant Manhattan" results pane is active, the left hand side results index takes a similar format as the gene Manhattan but displays single variant association p-values instead of gene test statistics. Single variant results can be filtered by consequence category (pLoF, missense, synonymous, and other). Clicking a variant ID will navigate to that variant's PheWAS view, and clicking the "details" arrow will keep the single variant Manhattan view active.

### Comparing single-variant associations across phenotypes

For a more comprehensive view of all variant-level associations for a gene in a single view, we developed functionality for exploring many phenotypes simultaneously on the gene PAGE (Figure S23). This feature aims to help users gain insight into pleiotropic patterns of variation across all high-scoring traits for a gene. Each row in the PheWAS table has a checkbox that, when checked, will overlay the phenotype in the gene plot (Figure S23A). The "select top" button will load all phenotypes below the $10^{-4}$ p-value threshold; the "clear selected" button will unselect all phenotypes and return to the single phenotype view (Figure S23B). When selected, phenotypes are assigned randomly generated colors to make them easier to distinguish in the plot and table (Figure S23C). Many tens or hundreds of phenotypes can be loaded simultaneously; however, an automatic p-value threshold will be applied when there are too many variants to display and the user will be warned in the single variant control panel.

By default, the variant table is configured to the "long" table format; when multiple phenotypes are selected, each variant-phenotype association will appear as a row in the variant table such that the variant table now contains duplicate entries for each variant. To make rows unique and to see comparison of association statistics across traits in a single row, the table can be set to "wide" format (Figure S24A). The phenotype pivots to the columns, creating a sort of genotype-phenotype matrix (Figure S24B). The column selection controls will affect both the long and wide table formats. When examining many phenotypes at once, users can click the "filter to selected" button on the phenotype section to simplify the PheWAS plot by only showing the selected phenotypes; this effectively serves as a legend for coloring-by-trait functionality (Figure S24C).

Hover interactions are especially useful when comparing multiple phenotypes; hovering over variants or phenotypes with the mouse will emphasize the relevant variants and bring them to the foreground (Figure S24D). The transparency slider sets the opacity level for non-hovered variants, helping the user tune the multi-phenotype plot such that hovered selections can stand out better (Figure S24E).

For categorical traits in particular, it is useful to get a visual sense of how case/control variant positions and allele frequencies differ across the gene. The "show case/control tracks" checkbox will fold out case/control tracks for all traits currently selected (Figure S25A). Continuous traits will be displayed in a single track and the allele frequency for individuals measured for the trait will be displayed. By viewing the case/control counts in the study (Figure S25B), alongside the case/control allele counts for variants in the variant table (Figure S25C) and the plots (Figure S25D) users can very quickly compare burden results across phenotypes, genes, and individual variants to get a sense of which specific variants may be driving gene burden signals. When the per-phenotype tracks are expended, it can be useful to use the "Color by" switch to look for trends and variants across genomic coordinate, trait, consequences, association p-value, effect size, and zygosity (Figure S26).

### Single variant results

When a variant is clicked on the single variant Manhattan plot or on the gene PAGE, the PAGE will focus on the selected variant, and a PheWAS displays all associations with that variant (Figure S27). Similar to the gene PheWAS PAGE, multiple phenotypes can be

selected and loaded at once. In the variant view, the table rows show statistics for the selected variant, and the columns show values across selected phenotypes. The variant position is displayed along the genomic coordinate. Clicking the "unselect" button will return to the gene PAGE. In this way, users can easily flip back and forth between single variants and the gene context.

## QUANTIFICATION AND STATISTICAL ANALYSIS

All statistical analysis was performed using SAIGE-GENE 0.44.2, R 4.0.2, and Hail 0.2.49-0.2.62. All methodological details can be found in the Method details, and all statistical tests are named as they are used.

## ADDITIONAL RESOURCES

All resulting summary statistics are publicly available as bulk downloads and in a browser interface at https://genebass.org.

