## [Document S2. Transparent peer review records for Karczewski et al · Cell Genomics]

Systematic single-variant and gene-based association testing of thousands of phenotypes in 426,370 UK Biobank exomes

Konrad J. Karczewski^{1,2,3,11,12}, Matthew Solomonson^{1,2,12}, Katherine R. Chao^{1,2,12}, Julia K. Goodrich^{1,2}, Grace Tiao^{1,2}, Wenhan Lu^{1,2,3}, Bridget M. Riley-Gillis⁴, Ellen A. Tsai⁵, Hye In Kim⁶, Xiuwen Zheng⁴, Fedik Rahimov⁴, Sahar Esmaeeli⁴, A. Jason Grundstad⁴, Mark Reppell⁴, Jeff Waring⁴, Howard Jacob⁴, David Sexton⁵, Paola G. Bronson⁵, Xing Chen⁶, Xinli Hu⁶, Jacqueline I. Goldstein^{1,2,3}, Daniel King^{1,2,3}, Christopher Vittal^{1,2,3}, Timothy Poterba^{1,2,3}, Duncan S. Palmer^{1,2,3}, Claire Churchhouse^{1,2,3}, Daniel P. Howrigan^{1,2,3}, Wei Zhou^{1,2}, Nicholas A. Watts^{1,2}, Kevin Nguyen^{1,2}, Huy Nguyen^{1,2}, Cara Mason⁷, Christopher Farnham⁷, Charlotte Tolonen⁷, Laura D. Gauthier⁷, Namrata Gupta⁷, Daniel G. MacArthur^{1,2,8,9}, Heidi L. Rehm^{1,2}, Cotton Seed^{1,2,3}, Anthony A. Philippakis⁷, Mark J. Daly^{1,2,3,10}, J. Wade Davis^{4,13}, Heiko Runz^{5,13}, Melissa R. Miller^{6,13}, Benjamin M. Neale^{1,2,3,14}

Summary

Initial submission: Received : November 8, 2021

Scientific editor: Orli Bahcall, Rosalind Mott and Laura Zahn

First round of review: Number of reviewers: 2
Revision invited : January 5, 2022
Revision received : March 20, 2022

Second round of review: Number of reviewers: 2
Accepted : July 16, 2022

Data freely available: Yes

Code freely available: Yes

This transparent peer review record is not systematically proofread, type-set, or edited. Special characters, formatting, and equations may fail to render properly. Standard procedural text within the editor's letters has been deleted for the sake of brevity, but all official correspondence specific to the manuscript has been preserved.

Referees' reports, first round of review

Referee#1:

I congratulate the authors for this is an important manuscript that should be of broad interest to the scientific community. I have the following mostly minor comments and suggestions that I hope will improve the manuscript.

Main text:

Did you have additional criteria beyond overall p-value for a group test to be considered significant? For example, did you require at least two variants in the corresponding mask?

Does SAIGE-GENE support analysis of related individuals? If so, why were close relatives excluded from the analysis? If not, please make that clear and justify the inclusion of more distant relatives.

If you are not analyzing the X chromosome, this should be made clear early on.

It is helpful when possible to include effect size estimates as well as pvalues. For example:

a. "we find that 470 genes previously implicated in developmental delay are more likely to be associated with a phenotype in the UK Biobank"

It would be helpful to say what the reader should do since you "urge caution in interpreting association results, particularly for the rarest binary traits (prevalence < 10⁻⁴) and for ultra-rare variants (frequency < 10⁻⁴)."

Figures

- a. 2AB: please describe in a bit more detail these very non-standard Manhattan plots
- b. 2CD: this is the first time categorical traits are mentioned; probably the term should be used in the text previously.
- c. 3: Please give the correlation values and significances in the text or figures.
- d. Several: I suspect I know what (0.1,) means, but this is non-standard. Same issue with (, 0.0001].
- e. 5: Many quantities are not defined. For example, panels should be labelled. What is No P-Val? What is Csq? AC trait? Should SCRIB be italicized?

Numbers:

- a. Were the 1,117 quantitative traits all such traits available on UKB participants, or was there some criterion for selecting them?
- b. Same question for the binary traits (beyond the requirement for ≥ 200 cases)?
- c. Similarly, how were the 19,591 protein-coding genes chosen?
- d. 57,650 is not equal to $3 \times 19,591$. Is this because some of the gene masks were identical or something else?
- e. How were the 7M single variants for testing chosen from the 20M detected?
- f. Please explain how 20,800,574,337 single variant association statistics result from 7,575,993 exome variants.
- g. It seems odd to claim 3700 phenotypes in the title and elsewhere when only 1362 passed QC.

Minor wording/punctuation:

- a. I am not sure how to parse "Using these criteria, we identified a total of 27,421, and 4,560 associations meeting our p-value threshold ...". Perhaps just delete the comma after 27,241?
- b. "this growth in effect size is slower than the loss in variance explained from their lower frequency" could

be interpreted to mean that effect size increases with time, which I believe is not your intent.

c. What are "autosomal dominant and autosomal recessive genes"? Genes with known single-gene diseases that are of these types?

d. Please explain what you mean by "the lack of asymptotic properties of the mixed-model tests for rarer binary traits"?

Methods:

There is a lot here, some of it simple, but much of it quite complex, consistent with the compute intensive nature of the project, and the strength of the team in this domain. The current version although long is also rather terse in multiple sections, with (for example) many terms used undefined. My own preference would be that these sections be extended to ensure used terms are defined. At the same time, I would understand if the authors prefer to stay largely where they are.

The Data Processing section is quite dense and rather difficult to follow.

The Scaling association testing using Hail Batch reads more like computer documentation than a scientific paper; this may be unavoidable.

p.34: Did you have an exclusion criterion based on concordance of sequence and array data? Or was 0.97 deemed acceptable?

pp. 51-53: Specification of the 314 models would be helpful. More important: given the observed heterogeneity across models observed below, some comment on the sufficiency/appropriateness of the 314 models chosen would be helpful.

p. 54: I do not understand the statement "To compute an effective number of tests and thus a p-value threshold for each phenotype, we took the minimum p-value for each of the simulated continuous phenotypes." Perhaps you could expand.

p. 55: The statement "Because of the highly deflated pattern of lambda GC values observed in genes with CAF 0-0.0001, we filtered out genes with CAF < 0.0001" seems odd to me. Given the observed deflation, I would have thought significant results with CAF < 0.0001 would be less likely to be false positives. Similar comment for the choice "we removed genes ... that have a synonymous lambda GC < 0.75."

Figures S11 and S12: Smaller points within the figures might make the pattern easier to see.

Figures S14 and S15: How was smoothing carried out for the figures?

p. 61: Did you filter variants with MAF < 0.01% as perhaps suggested by Table S2? If so, I missed this and do not find it when re-reading. Explanation should be provided if it is not. If there is reason to do this, it would raise the question of whether sequencing is a (near) waste of time compared to array-genotyping and imputation since we can get to fairly high imputation accuracy for variants with MAF>0.01% for populations well represented in imputation reference panels. Also, why are annotation defined entries not provided in the table except for all variants?

p. 61: Are the single-variant results directionally consistent with past findings? If there are burden test results, are they directionally consistent?

p. 72: In your section on "Gene set analyses: Developmental delay genes", I assume you used the existing genotypes and phenotypes. This should probably be clarified. Also, it would be nice to include a measure of enrichment rather than just the pvalue.

Minor wording/punctuation:

Please reduce the use of the passive voice, since it makes for less interesting reading, and more importantly, makes it less clear when you did the work.

p. 31: Please explain the relevance of "three rounds of merging". This is apparently the only place in the text in which the idea of rounds of merging is used.

p. 33, 77: "data are" not "data is"

p. 38: "a second-degree or greater relationship" would be more clear as "a second-degree or closer relationship" if that is what you mean

p. 50 and elsewhere: "data were" not "data was"

"In order to" can almost always be replaced by "To"

"heritable phenotypes" are apparently phenotypes with heritability 1. This non-standard terminology should be explicitly defined or perhaps better yet, not used.

p. 51: Figure S9: What is "additional heritability fraction"?

p. 77, 80, and later: capitalize Manhattan?

p. 79: "The plot transitions from to a SINGLE LOG SCALE TO A double log scale $\frac{2}{3}$ along the plot height"; that is, add the capitalized text.

p. 81: "the three burden test types are displays as columns"; displays should be displayed?

Referee#2:

The authors present here their work on rare-variant analyses in 300k exomes from the UK Biobank and introduce the Genebass resource. The manuscript is concise and well-written, although quite light on interesting association results. The main highlight is the introduction of Genebass, which should be a phenomenal resource for geneticists and non-geneticists alike for exploration of association results from biobank-scale sequencing datasets.

Mostly minor comments and questions to address:

The introduction, particularly the 3rd paragraph, is a bit disjointed. I would suggest revising.

The UK Biobank 450k WES dataset was recently released (see Backman et al, Nature 2021). Please comment on whether readers should expect to see an update to Genebass with 450k WES and/or WGS?

Did authors consider gene-based tests that combine across functional categories? Since you are utilizing SCAT/SCAT-O and directionality is not an important consideration, it would seem a reasonable way to boost power.

I am not sure I fully understand the empirical p-value threshold estimation. Why were 314 heritable traits chosen for simulation? Was correlation amongst phenotypes considered in the model? A mathematical description of this model might be more clear. This section is not clear: "Thus, for downstream analyses, we computed the experiment-wise significance threshold..."; Are authors dividing by 20 to get Type 1 error at 5%? Is 5×10^{-6} a typo? $20 * 2.5 \times 10^{-8} = 5 \times 10^{-7}$. How does this threshold compare to a strict Bonferroni?

For the gene-burden test, is there a maximum allele frequency for variants to be included? If not, it is quite likely that many associations are driven by nearby common variants.

Can authors briefly comment on why these 3 particular gene-based tests were chosen?

Considering differences in sample size across phenotypes, why did authors choose a single MAF/cMAF cutoff? Would a lower-bound defined by minor allele count or cMAC not be more appropriate?

I am quite surprised many more associations are observed for single variants than for gene-based tests, but I presume this is due to associations with multiple variants annotated to the same gene. How many unique gene-trait pairs are observed? Similarly, of the 1,069 gene-based tests in which no significant single-variant association was observed, how many unique gene-trait pairs? Would be informative to also see these broken down into functional category. Further, how many of these are known or are in genes with known GWAS associations for the same/similar trait?

Was a comparison between Burden v SKAT v SKAT-O completed for each functional category? For example, did Burden outperform SKAT for pLOF aggregate tests, where one would hypothesize same direction of effect, whereas the missense aggregate test may include both LOF and GOF?

Figure 3A. The deleteriousness pattern is not observable at the 2 lowest frequency groupings. You may want to change the scale on the Y-axis. Additionally, could you add commentary in the text why the pattern does not hold in the most common (0.1,) group?

Consistent with the polyphen2 observation, did authors consider gene-based test of possibly/probably deleterious missense variants?

The authors note that constrained genes are more likely to be associated with a phenotype than a frequency-matched set of genes in the genome. Are the observed associations with constrained genes fairly specific to a subset of phenotypes (eg. Neurodevelopmental) or are these associations quite diverse?

Can the authors comment on the lack of association for constrained genes in the missense category in Figure 4? Would an association be observed if restricted to polyphen pathogenic/likely pathogenic?

Genebase is very well-designed, useful resource. One could spend hours digging around for their favorite gene and fall deep into a rabbit-hole of interesting results. Are there plans to make the code open-source for others to clone and implement with their own sequence data?

One key feature of Genebase is ability to evaluate pleiotropy. Can authors comment in the main text on this? Avg # of associated phenotypes per gene, for example.

I think the SCRIB story in the biological insights section is interesting, but there is a missed opportunity here to highlight pleiotropy, which is one of the most exciting aspects of the Genebase browser. The authors note a mean of 20.1 and 3.35 associations per phenotype, for single-variant tests and group tests, respectively. Some notable examples that highlight the utility of Genebase for novel discovery would be very nice to include here. Similarly, examples of allelic series could be quite compelling.

Fig. S1: For 3332, shouldn't LA=[0] in the merged, as seen for 3330? For 3350 sample 2 in the merged, shouldn't this be LA=[0,2]. This part is not clear to me.

For the RF model, in addition to balancing across training sets, do you balance across allele frequencies? Have you observed any biases in pass/fail rate across frequency spectra?

Could authors explain why leave one chromosome out was not used for SAIGE?

In selecting independent phenotypes, largest N seems reasonable given the # of traits tested. Can authors comment on whether a more nuanced approach, such as selecting the trait with highest h^2 , might offer better power in some cases?

Authors mention in supplement that associations were filtered due to a bug in SAIGE-GENE. Will this be corrected and added back in?

One limitation not mentioned is no accounting for common variants. It is plausible that some associations observed here are in LD/partial LD with common variants not tested.

Authors' response to the first round of review

Reviewer #1:

I congratulate the authors for this is an important manuscript that should be of broad interest to the scientific community. I have the following mostly minor comments and suggestions that I hope will improve the manuscript.

We are glad the reviewer found the manuscript of interest and thank them for their helpful comments. We have incorporated the comments and believe this has improved the manuscript.

Main text:

Did you have additional criteria beyond overall p-value for a group test to be considered significant? For example, did you require at least two variants in the corresponding mask?

Thank you for this question. We indeed required at least two variants, which was noted deep in the supplement. We have now included this in the main text that the QC also includes "a minimum of two variants per group test, a minimum coverage of 20X, [...]".

Does SAIGE-GENE support analysis of related individuals? If so, why were close relatives excluded from the analysis? If not, please make that clear and justify the inclusion of more distant relatives.

Indeed, SAIGE-GENE supports analysis of related individuals, and we included these in our analysis. Close relatives were excluded for the purposes of PCA for ancestry inference, but were re-introduced (after projecting PCs to infer ancestry for the related individuals). We have made this clear in the manuscript. If you are not analyzing the X chromosome, this should be made clear early on.

The summary statistics were generated for all chromosomes, including X and Y, and analyses performed across all assayed genes.

It is helpful when possible to include effect size estimates as well as p-values. For example:

a. "we find that 470 genes previously implicated in developmental delay are more likely to be associated with a phenotype in the UK Biobank"

We have edited the text to include effect sizes (odds ratios around the Fisher's test enrichments).

"In particular, we find that 470 genes previously implicated in developmental delay (Kaplanis et al., 2020) are more likely to be associated with a phenotype in the UK Biobank (Fisher's exact $p = 3.6 \times 10^{-4}$, OR = 3.50; Fig. 4). Further, we observe a correlation between selection against pLoFs in a gene and the phenotypic impact of pLoFs in that gene: specifically, constrained genes (i.e., those in the highest decile of LoF observed/expected upper bound fraction [LOEUF], a metric of LoF intolerance) are more likely to be associated with a phenotype (9.14%) than a frequency-matched set of genes in the genome (2.12%; Fisher's exact $p = 6.1 \times 10^{-14}$, OR = 4.65; Fig. 4)."

It would be helpful to say what the reader should do since you "urge caution in interpreting association results, particularly for the rarest binary traits (prevalence $< 10^{-4}$) and for ultra-rare variants (frequency $< 10^{-4}$)."

We have clarified this section with: "For the rarest outcomes, increasing the number of cases is essential to properly evaluate the impact of rare coding variation across genes. Alternatively, other statistical methods such as Firth regression may be better suited to such traits. For pLoF variants, the median cumulative allele frequency across genes is approximately 1.5×10^{-4} , suggesting that group tests at current sample sizes are only powered to detect individual gene effects for quantitative traits that capture at least 0.02% of variance, as well as diseases and traits that have a high prevalence (well above 10%; Fig. S10)."

Figures

a. 2AB: please describe in a bit more detail these very non-standard Manhattan plots

We have clarified this further "For each ICD chapter, we show a Manhattan plot, depicting the distribution of p-values for all single-variant (A) and SKAT-O gene-based (B) associations, where for each variant/gene, the minimum p-value across phenotypes within each category is shown."

b. 2CD: this is the first time categorical traits are mentioned; probably the term should be used in the text previously.

We have edited this in the figure to "binary" to be consistent with the remainder of the manuscript. In the supplement, we maintain both descriptors, as the browser is built around "categorical" so we clarify that 'For this supplement, we use the descriptors "categorical" and "binary" interchangeably when describing phenotypes.'

c. 3: Please give the correlation values and significances in the text or figures.

For Figure 4, which has significance values shown, these were the result of permutation tests, which have only p-values and not effect sizes.

d. Several: I suspect I know what (0.1,) means, but this is non-standard. Same issue with (, 0.0001].

We have edited this to be (0.1, ∞) and [0, 0.0001].

e. 5: Many quantities are not defined. For example, panels should be labelled. What is No

P-Val? What is Csq? AC trait? Should SCRIB be italicized?

We have fixed this figure to label the panels, define abbreviations, and italicize SCRIB.

Numbers:

a. Were the 1,117 quantitative traits all such traits available on UKB participants, or was there some criterion for selecting them?

b. Same question for the binary traits (beyond the requirement for ≥ 200 cases)?

The revised manuscript now includes 1,233 quantitative and 3,296 binary traits that were selected from continuous and categorical data fields as provided by UK Biobank (<https://biobank.ndph.ox.ac.uk/showcase/list.cgi>). In addition to a requirement for a minimum of 200 cases, traits were further prioritized semi-manually based on trait redundancy and assumed relevance to disease. A subset of custom binary traits are manually curated composite endpoints that based on expert input can be expected to more accurately reflect the respective disease categories than ICD10 codes alone.

c. Similarly, how were the 19,591 protein-coding genes chosen?

These were all genes that had at least one variant identified in our dataset from the UKB exome capture platform.

d. 57,650 is not equal to $3 \times 19,591$. Is this because some of the gene masks were identical or something else?

The bulk of this discrepancy comes from the pLoF group, as not all genes have any pLoF variants due to selection or gene size. The full breakdown is provided in Table S2.

e. How were the 7M single variants for testing chosen from the 20M detected?

Similarly, the 7 million variants are only pLoF (stop-gained, essential splice, and frameshift), missense, or synonymous. Exon-adjacent variants such as extended splice or intronic variants may have been called in the initial callset but not included in the association analysis. We have clarified that this filter is the reason for the discrepancy.

f. Please explain how 20,800,574,337 single variant association statistics result from 7,575,993 exome variants.

This number refers to the total number of tests performed across phenotypes

g. It seems odd to claim 3700 phenotypes in the title and elsewhere when only 1362 passed QC.

We initially used this number as this was the number of tests performed, and we have released all the results to the community. However, as it can be misleading, we have edited the title to "thousands of phenotypes"

Minor wording/punctuation:

a. I am not sure how to parse "Using these criteria, we identified a total of 27,421, and 4,560 associations meeting our p-value threshold ...". Perhaps just delete the comma after 27,241?

b. "this growth in effect size is slower than the loss in variance explained from their lower frequency" could be interpreted to mean that effect size increases with time, which I believe is not your intent.

c. What are "autosomal dominant and autosomal recessive genes"? Genes with known single-gene diseases that are of these types?

d. Please explain what you mean by "the lack of asymptotic properties of the mixed-model tests for rarer binary traits"?

We have incorporated all these comments and clarified the text around all these.

Methods:

There is a lot here, some of it simple, but much of it quite complex, consistent with the compute intensive nature of the project, and the strength of the team in this domain. The current version although long is also rather terse in multiple sections, with (for example) many terms used undefined.

My own preference would be that these sections be extended to ensure used terms are defined. At the same time, I would understand if the authors prefer to stay largely where they are.

Indeed the supplement is quite long as it represents the documentation of all the steps we performed. We opted to cite previous work where possible in order to keep it as short as possible to focus on the novel aspects (sparse variant representation, a computational framework for large-scale analysis, and the novel quality control metrics for rare variant associations).

The Data Processing section is quite dense and rather difficult to follow.

We have edited this text to hopefully make it more readable and a bit easier to follow.

The Scaling association testing using Hail Batch reads more like computer documentation than a scientific paper; this may be unavoidable.

We do believe this is somewhat unavoidable as this is describing the technical details of a quite complicated computational system.

p.34: Did you have an exclusion criterion based on concordance of sequence and array data? Or was 0.97 deemed acceptable?

Indeed, this analysis was meant primarily to identify sample swaps, and we were prepared to exclude samples based on concordance, but after observing the distribution of concordance, we deemed that a minimum of 96% was sufficiently high, with the mean being 99.5%.

pp. 51-53: Specification of the 314 models would be helpful. More important: given the observed heterogeneity across models observed below, some comment on the sufficiency/appropriateness of the 314 models chosen would be helpful.

We have clarified that these are random normally distributed phenotypes. We note that the lambda values for these phenotypes (Figure S9) closely resemble those of real phenotypes at similar prevalences (Figure S11), so we believe these models are representative of the asymptotic properties of the statistical analysis of the data.

p. 54: I do not understand the statement "To compute an effective number of tests and thus a p-value threshold for each phenotype, we took the minimum p-value for each of the simulated continuous phenotypes." Perhaps you could expand.

We are happy to clarify this. We ran our association testing suite (SAIGE-GENE) between the genotype data and these randomly simulated phenotypes. The most significant p-value is an estimator for the number of effective tests, so we take $0.05 * p$ as our p-value threshold. We have clarified this in the supplement.

p. 55: The statement "Because of the highly deflated pattern of lambda GC values observed in genes with CAF 0-0.0001, we filtered out genes with CAF < 0.0001" seems odd to me. Given the observed deflation, I would have thought significant results with CAF < 0.0001 would be less likely to be false positives. Similar comment for the choice "we removed genes ... that have a synonymous lambda GC < 0.75."

Indeed these are unlikely to be Type II errors. However, as we wanted to balance the number of type I errors in our analysis across genes and gene sets, we opted to remove these from our analyses. We provide the full set of association statistics in the data release so that these can be recovered if desired although we remain concerned about the stability of the association test in circumstances with few cases and few alternate alleles.

Figures S11 and S12: Smaller points within the figures might make the pattern easier to see.

Thank you for the suggestion, we have made the points smaller in these figures.

Figures S14 and S15: How was smoothing carried out for the figures?

We have clarified that these are standard density plots.

p. 61: Did you filter variants with MAF < 0.01% as perhaps suggested by Table S2? If so, I missed this and do not find it when re-reading. Explanation should be provided if it is not. If there is reason to do this, it would raise the question of whether sequencing is a (near) waste of time compared to array-genotyping

and imputation since we can get to fairly high imputation accuracy for variants with MAF>0.01% for populations well represented in imputation reference panels. Also, why are annotation defined entries not provided in the table except for all variants?

This is noted in the text where we filter to “variants with a cohort frequency of at least 0.01%”. Variants with MAF < 0.01% are filtered out from analysis of single-variant associations as they are individually underpowered, but not from the gene-based tests as they can be aggregated to increase power in the burden/SKAT framework. The “annotation defined” refers to a set of all but 4,214 (now 5,824) variants that have a pLoF, missense, or synonymous annotation, and as such the numbers are the same as above. We have copied the numbers for simplicity.

p. 61: Are the single-variant results directionally consistent with past findings? If there are burden test results, are they directionally consistent?

The single variant results are directionally consistent and consistent in magnitude with past findings. We show a comparison to GIANT statistics for height in Supplementary Figure 17, and there is a good correlation with some attenuation as expected due to winner’s curse. For burden test results, we find a high degree of replication (Table S5); however, burden betas were not available as GIANT used SKAT and VT, and do not provide effect sizes in (Marouli et al 2017).

p. 72: In your section on “Gene set analyses: Developmental delay genes”, I assume you used the existing genotypes and phenotypes. This should probably be clarified. Also, it would be nice to include a measure of enrichment rather than just the pvalue.

Indeed, we used a set of previously-established developmental delay genes, and assessed the enrichment of associations in the Genebase dataset among these genes. We have added the odds ratio to the main text.

Minor wording/punctuation:

Please reduce the use of the passive voice, since it makes for less interesting reading, and more importantly, makes it less clear when you did the work.

We have edited the supplement extensively to minimize the passive voice.

p. 31: Please explain the relevance of “three rounds of merging”. This is apparently the only place in the text in which the idea of rounds of merging is used.

We have clarified that the paragraph above is run in several rounds for a hierarchical merge: “repeating the second step in rounds until a single SparseMT is created.”

p. 33, 77: “data are” not “data is”

Fixed.

p. 38: “a second-degree or greater relationship” would be more clear as “a second-degree or closer relationship” if that is what you mean

Fixed.

p. 50 and elsewhere: “data were” not “data was”

Fixed.

“In order to” can almost always be replaced by “To”

Fixed.

“heritable phenotypes” are apparently phenotypes with heritability 1. This non-standard terminology should be explicitly defined or perhaps better yet, not used.

We have clarified this - we generated heritability 1 phenotypes as well as phenotypes with a range of heritabilities.

p. 51: Figure S9: What is “additional heritability fraction”?

This has been clarified.

p. 77, 80, and later: capitalize Manhattan?

Fixed.

p. 79: "The plot transitions from to a SINGLE LOG SCALE TO A double log scale $\frac{2}{3}$ along the plot height"; that is, add the capitalized text.

Fixed.

p. 81: "the three burden test types are displays as columns"; displays should be displayed?

Fixed.

Reviewer #2: The authors present here their work on rare-variant analyses in 300k exomes from the UK Biobank and introduce the Genebass resource. The manuscript is concise and well-written, although quite light on interesting association results. The main highlight is the introduction of Genebass, which should be a phenomenal resource for geneticists and non-geneticists alike for exploration of association results from biobank-scale sequencing datasets.

We are grateful for the reviewer's comments and are glad that they find the resource useful. We have addressed their comments and believe they have improved the manuscript.

Mostly minor comments and questions to address:

The introduction, particularly the 3rd paragraph, is a bit disjointed. I would suggest revising.

We have revised the third paragraph, including additions that address the other publications of the exome resource in the meantime.

The UK Biobank 450k WES dataset was recently released (see Backman et al, Nature 2021). Please comment on whether readers should expect to see an update to Genebass with 450k WES and/or WGS?

We are as excited as this reviewer to have the latest and greatest dataset, and indeed, in the interim, we were able to run the association tests for the 450k WES dataset. We have updated the entire manuscript to reflect the final dataset. We do not have plans at this time to use the WGS data, as it is not obvious how non-coding regions should be tested, but this is a potential topic for future work.

Did authors consider gene-based tests that combine across functional categories? Since you are utilizing SCAT/SCAT-O and directionality is not an important consideration, it would seem a reasonable way to boost power.

In the original 300k version of the manuscript, we did not combine across functional categories.

However, in the 450k dataset, we have also run combined tests of missense + pLoF variants. We have added a note around this result in the text:

"In combined tests of pLoF and missense variants, we find an additional 267 associations among burden tests (245 for SKAT-O) that are significant for the combined test but not missense or pLoF tests alone."

I am not sure I fully understand the empirical p-value threshold estimation. Why were 314 heritable traits chosen for simulation? Was correlation amongst phenotypes considered in the model? A mathematical description of this model might be more clear. This section is not clear:

"Thus, for downstream analyses, we computed the experiment-wise significance threshold..."; Are authors dividing by 20 to get Type 1 error at 5%?

We appreciate the opportunity to clarify, these were not chosen from the existing phenotypes but newly generated random normal distributions to simulate noise phenotypes. There were 314 of these, due to replicates, as well as a range of types (quantitative and binary), prevalences (for binary traits), and heritabilities (the genetic relatedness matrix was used in the generation of the phenotypes to introduce heritability). We then ran our association testing suite (SAIGE-GENE) between the genotype data and these randomly simulated phenotypes. The most significant p-value is an estimator for the number of effective tests, so we take $0.05 * p$ as our p-value threshold. We have clarified this in the supplement.

Is 5×10^{-6} a typo? $20 * 2.5 \times 10^{-8} = 5 \times 10^{-7}$. How does this threshold compare to a strict Bonferroni?

We are very appreciative of this catch - indeed, there is a typo, but it is in the p-value threshold! 5×10^{-6} was indeed the median minimum p-value across the heritable continuous traits, so the p-value threshold should have been 2.5×10^{-7} . We have now fixed this in the text and updated all numbers

throughout to reflect this updated threshold. For group tests, this was somewhat consistent with Bonferroni, as $0.05/76767 = 6.6 \times 10^{-7}$, which is similar to the burden threshold and within a factor of 3 of the new SKAT-O threshold. For single-variant tests, this was also similar ($0.05/8074878 = 6.2 \times 10^{-9}$) compared to 8×10^{-9} .

For the gene-burden test, is there a maximum allele frequency for variants to be included? If not, it is quite likely that many associations are driven by nearby common variants.

Indeed, this was a concern in our analysis. SAIGE downweights common variants by a Beta(1,25) distribution, such that in practice, variants at >5% frequency are several-fold decreased in weight compared to singletons. To address this, in our update to the 450k, we have strictly removed variants at >1% frequency from the burden test, and this has not materially changed the results. We have updated all results with these variants removed in the new version of the manuscript.

Can authors briefly comment on why these 3 particular gene-based tests were chosen?

These are the three gene-based tests implemented in SAIGE-GENE. We wanted a mean-based test (Burden) and a variance test (SKAT), and the hybrid approach (SKAT-O) was a computationally efficient addition to these.

Considering differences in sample size across phenotypes, why did authors choose a single MAF/cMAF cutoff? Would a lower-bound defined by minor allele count or cMAC not be more appropriate?

We thank the reviewer for this idea, as the statistics are conditional on both the allele frequency and prevalence of the disease. We have changed our filtering process to retain tests where $MAF * n_cases$ (for quantitative traits, the total sample size) ≥ 50 , and similar for gene-based tests ($CAF * n_cases \geq 50$). We have updated the numbers throughout the paper to reflect this change.

I am quite surprised many more associations are observed for single variants than for gene-based tests, but I presume this is due to associations with multiple variants annotated to the same gene. How many unique gene-trait pairs are observed? Similarly, of the 1,069 gene-based tests in which no significant single-variant association was observed, how many unique gene-trait pairs?

We appreciate the opportunity to clarify this point, as we were primarily highlighting the associations gained by group tests rather than providing a direct comparison between the single-variant and group tests. The results of this analysis are already collapsed into unique gene-trait pairs, and indeed, the single-variant tests identify more associations. As the reviewer notes, associations with multiple variants annotated to the same gene is likely to occur because of linkage disequilibrium. We have clarified the point about more associations for single-variant tests in the text:

“Comparing the group test results to single-variant association test results, we find that single-variant tests identify more significant associations than group tests, as these are largely from common variants that are excluded from the group tests. However, we also find 2,237 associations (on average 0.62 per phenotype) from group tests where no single-variant association reached our p-value threshold for any single variant in the corresponding gene (Fig. 2C).”

Would be informative to also see these broken down into functional category. Further, how many of these are known or are in genes with known GWAS associations for the same/similar trait?

We thank the reviewer for this comment: we have explored the breakdown of associations by functional category and the result is very interesting, so we are adding it as Figure 2D. As might be expected due to their larger number, missense and synonymous variants show the most associations, but we observe a higher fraction of associations for gene-based tests for pLoF variants, which is consistent with these variants being individually rare, but their directional and biological consistency as well as the higher confidence in functional impact provides additional power when incorporating into a group test. We have not performed a systematic comparison to known GWAS associations as this would require accurate matching of phenotypes: we are building this infrastructure and will incorporate it into future work.

Was a comparison between Burden v SKAT v SKAT-O completed for each functional category?

For example, did Burden outperform SKAT for pLoF aggregate tests, where one would hypothesize same direction of effect, whereas the missense aggregate test may include both LOF and GOF?

We thank the reviewer for the question as although we do not have gold standard data to assess which regime would have better performance, we did explore the properties of the significant associations across these analyses. We found that for pLoF variants, there is a greater relative ratio of significant hits for the burden test (1,274) over the SKAT-O (1,628), compared to missense variants (1,281 for burden, 2,256 for SKAT-O), suggesting that this hypothesis is plausible, and that variance-based tests may be more robust to a larger number of benign variants.

Figure 3A. The deleteriousness pattern is not observable at the 2 lowest frequency groupings. You may want to change the scale on the Y-axis. Additionally, could you add commentary in the text why the pattern does not hold in the most common (0.1,) group?

We have modified this figure to highlight the pattern in the lower groups, by moving the higher group to the supplement on a separate scale. We have also added the commentary around the common groups: “For common variants (>1%), we observe further increases in associations due to power, but with attenuated associations for pLoF variants, likely due to an increased rate of artifacts at common pLoF variants (MacArthur and Tyler-Smith, 2010) (Fig. S18).”

Consistent with the polyphen2 observation, did authors consider gene-based test of possibly/probably deleterious missense variants?

We appreciate the suggestion and also believe this would be an interesting avenue for exploration. However, the current implementation of SAIGE-GENE requires an a priori assignment of groups and each group (particularly one with lots of variants like missense) adds substantially to the cost of analysis. A meta-analysis method such as ACAT-V may be more suited to exploration of several groupings of gene-based tests.

The authors note that constrained genes are more likely to be associated with a phenotype than a frequency-matched set of genes in the genome. Are the observed associations with constrained genes fairly specific to a subset of phenotypes (eg. Neurodevelopmental) or are these associations quite diverse?

We thank the reviewer for another interesting question to pursue. We explored how this result would break down across phenotype categories. We compared the proportion of constrained genes with associations with categorical, continuous, and disease phenotypes to a frequency-matched set of genes (see Figure below). The majority of the associations discovered were mostly driven by the continuous phenotypes, and so, there was not enough power to detect the differences for categorical or disease traits, let alone a specific subgroup of diseases, given the relatively small number of associations. Within the continuous traits, we observe physical measures and touchscreen questionnaire responses as enriched (below), but we do not have a further classification to explore subsets at this time.

Can the authors

comment on the lack of association for constrained genes in the missense category in Figure 4? Would an association be observed if restricted to polyphen pathogenic/likely pathogenic?

The effect of missense variants is more heterogeneous, as evidenced by only a light correlation between missense constraint and pLoF constraint. It is possible that running group tests for likely pathogenic variants (or missense variants in missense-constrained regions) might hone the signal, but we did not do this due to computational complexity.

Genebase is very well-designed, useful resource. One could spend hours digging around for their favorite gene and fall deep into a rabbit-hole of interesting results. Are there plans to make the code open-source for others to clone and implement with their own sequence data?

We indeed have plans to make this code open source. We are currently preparing a larger software suite for serving association results that we plan to release when the code has stabilized and the installation protocol is well documented.

One key feature of Genebase is ability to evaluate pleiotropy. Can authors comment in the main text on this? Avg # of associated phenotypes per gene, for example.

I think the SCRIB story in the biological insights section is interesting, but there is a missed opportunity here to highlight pleiotropy, which is one of the most exciting aspects of the Genebase browser. The authors note a mean of 20.1 and 3.35 associations per phenotype, for single-variant tests and group tests, respectively. Some notable examples that highlight the utility of Genebase for novel discovery would be very nice to include here. Similarly, examples of allelic series could be quite compelling.

We agree that these are very interesting avenues for exploration. We are currently working on a separate manuscript on pleiotropy and allelic series which will cover these statistics and more. We note this in the Discussion: "Future work will be needed to fully assess the contribution of rare variants to the heritability of common diseases, as well as the extent and role of pleiotropy among rare variants."

Fig. S1: For 3332, shouldn't LA=[0] in the merged, as seen for 3330? For 3350 sample 2 in the merged, shouldn't this be LA=[0,2]. This part is not clear to me.

Thank you for this catch. 3350 sample 2 should definitely be LA=[0,2], and for reference blocks, LA=[0] is implied so that was removed. These and other fixes to this figure have been made.

For the RF model, in addition to balancing across training sets, do you balance across allele frequencies? Have you observed any biases in pass/fail rate across frequency spectra?

We have typically not balanced across allele frequencies. Most training data has historically been common variants (HapMap, Omni, etc) and we have recently introduced transmitted and sibling singletons to fill the rare end of the spectrum. We had previously tried to titrate the relative numbers of common and rare variants in gnomAD, but at the time, it was not clear which balance was superior: further work would be warranted to assess this more systematically.

Could authors explain why leave one chromosome out was not used for SAIGE?

We did not use the LOCO version of SAIGE for computational efficiency (the step1 is several-fold more expensive when running LOCO).

In selecting independent phenotypes, largest N seems reasonable given the # of traits tested. Can authors comment on whether a more nuanced approach, such as selecting the trait with highest h^2 , might offer better power in some cases?

Indeed, we've thought about this extensively and we believe there are probably multiple valid approaches. From a practical perspective, we have not computed h^2 values for this dataset, as methods to do so for rare variants do not exist. It would in principle be possible to cross-reference to common variant GWAS results, but these have not been performed on the same set of phenotypes, and so we wanted a complete and unbiased approach which we found with largest N.

Authors mention in supplement that associations were filtered due to a bug in SAIGE-GENE. Will this be corrected and added back in?

Unfortunately, this bug was fixed after the analysis was complete, and it would be computationally challenging to redo just these associations. As it is a small handful we do not believe it will affect the results of the analyses.

One limitation not mentioned is no accounting for common variants. It is plausible that some associations observed here are in LD/partial LD with common variants not tested.

We agree that LD could affect some of these results. To assuage this concern, in the new update to the 450K, we have removed variants at >1% frequency from the burden analysis. In the future, we plan to compute LD between all pairs of variants within each gene in the dataset to more formally assess the effect of LD; however, as this is quite computationally expensive, we are building the infrastructure to do so.

Referees' report, second round of review

Referee#1:

The authors have thoughtfully addressed my comments and questions. Congratulations on an excellent paper!

Referee#2:

Congratulations on a wonderful paper and resource!

Authors' response to the second round of review